# Imprinted SARS-CoV-2 humoral immunity induces convergent Omicron RBD evolution

Yunlong Cao[1,2,11✉], Fanchong Jian[1,3,11], Jing Wang[1,4,11], Yuanling Yu[2,11], Weiliang Song[1,4,11], Ayijiang Yisimayi[1,4], Jing Wang[2], Ran An[2], Xiaosu Chen[5], Na Zhang[2], Yao Wang[2], Peng Wang[2], Lijuan Zhao[2], Haiyan Sun[2], Lingling Yu[2], Sijie Yang[1,6], Xiao Niu[1,3], Tianhe Xiao[1,7], Qingqing Gu[2], Fei Shao[2], Xiaohua Hao[8], Yanli Xu[8], Ronghua Jin[8], Zhongyang Shen[9], Youchun Wang[2,10✉] & Xiaoliang Sunney Xie[1,2✉]

Continuous evolution of Omicron has led to a rapid and simultaneous emergence of numerous variants that display growth advantages over BA.5 (ref. [1]). Despite their divergent evolutionary courses, mutations on their receptor-binding domain (RBD) converge on several hotspots. The driving force and destination of such sudden convergent evolution and its effect on humoral immunity remain unclear. Here we demonstrate that these convergent mutations can cause evasion of neutralizing antibody drugs and convalescent plasma, including those from BA.5 breakthrough infection, while maintaining sufficient ACE2-binding capability. BQ.1.1.10 (BQ.1.1 + Y144del), BA.4.6.3, XBB and CH.1.1 are the most antibody-evasive strains tested. To delineate the origin of the convergent evolution, we determined the escape mutation profiles and neutralization activity of monoclonal antibodies isolated from individuals who had BA.2 and BA.5 breakthrough infections[2,3]. Owing to humoral immune imprinting, BA.2 and especially BA.5 breakthrough infection reduced the diversity of the neutralizing antibody binding sites and increased proportions of non-neutralizing antibody clones, which, in turn, focused humoral immune pressure and promoted convergent evolution in the RBD. Moreover, we show that the convergent RBD mutations could be accurately inferred by deep mutational scanning profiles[4,5], and the evolution trends of BA.2.75 and BA.5 subvariants could be well foreseen through constructed convergent pseudovirus mutants. These results suggest that current herd immunity and BA.5 vaccine boosters may not efficiently prevent the infection of Omicron convergent variants.

SARS-CoV-2 Omicron BA.1, BA.2 and BA.5 have demonstrated strong neutralization evasion capability, posing severe challenges to the efficacy of existing humoral immunity established through vaccination and infection[2,3,6–15]. Nevertheless, Omicron is continuously evolving, leading to various new subvariants, including BA.2.75, BA.4.6 and BF.7 (refs. [16–21]). A high proportion of these emerging variants display substantial growth advantages over BA.5, such as BA.2.3.20, BA.2.75.2, BQ.1.1 and especially XBB, a recombinant of BJ.1 and BM.1.1.1 (ref. [1]) (Fig. 1). Such rapid and simultaneous emergence of multiple variants with enormous growth advantages is unprecedented. Of note, although these derivative subvariants appear to diverge along the evolutionary course, the mutations they carry on the RBD converge on the same sites, including R346, K356, K444, V445, G446, N450, L452, N460, F486, F490, R493 and S494 (Fig. 1). These residues were found mutated in at least five independent

Omicron sublineages that exhibited a high growth advantage (Extended Data Fig. 1a–c). Most mutations on these residues are known to be antibody-evasive, as revealed by deep mutational scanning (DMS)[2,3,22–24]. It is crucial to examine the effect of these convergent mutations on antibody-escaping capability, receptor-binding affinity, and the efficacy of vaccines and antibody therapeutics. It is also important to investigate the driving force behind this suddenly accelerated emergence of convergent RBD mutations, what such mutational convergence would lead to and how we can prepare for such rapid SARS-CoV-2 evolution.

## Antibody evasion by convergent variants

First, we tested the antibody evasion capability of these convergent variants. We constructed the vesicular stomatitis virus (VSV)-based

[1]Biomedical Pioneering Innovation Center (BIOPIC), Peking University, Beijing, P. R. China. [2]Changping Laboratory, Beijing, P. R. China. [3]College of Chemistry and Molecular Engineering, Peking University, Beijing, P.R. China. [4]School of Life Sciences, Peking University, Beijing, P.R. China. [5]Institute for Immunology, College of Life Sciences, Nankai University, Tianjin, P. R. China. [6]Peking-Tsinghua Center for Life Sciences, Peking University, Beijing, P. R. China. [7]Joint Graduate Program of Peking-Tsinghua-NIBS, Academy for Advanced Interdisciplinary Studies, Peking University, Beijing, China. [8]Beijing Ditan Hospital, Capital Medical University, Beijing, P. R. China. [9]Organ Transplant Center, NHC Key Laboratory for Critical Care Medicine, Tianjin First Central Hospital, Nankai University, Tianjin, P. R. China. [10]Division of HIV/AIDS and Sex-transmitted Virus Vaccines, Institute for Biological Product Control, National Institutes for Food and Drug Control (NIFDC), Beijing, P. R. China. [11]These authors contributed equally: Yunlong Cao, Fanchong Jian, Jing Wang, Yuanling Yu, Weiliang Song. ✉e-mail: yunlongcao@pku.edu.cn; wangyc@nifdc.org.cn; sunneyxie@biopic.pku.edu.cn

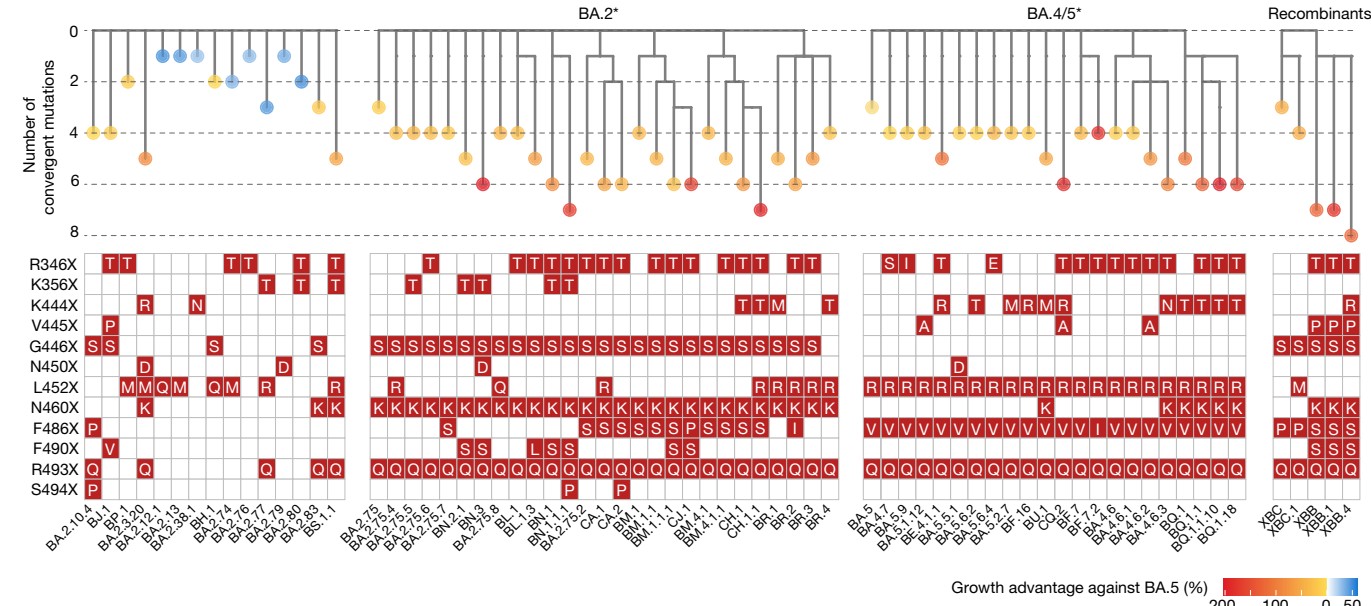

**Fig. 1 | Convergent evolution of Omicron RBD with growth advantage over BA.5.** Phylogenetic tree of featured Omicron subvariants carrying convergent mutations. Their relative growth advantage values, calculated using the CoV-Spectrum website, are indicated as a colour scale. Specific convergent mutations carried by each lineage are labelled.

spike-pseudotyped virus of Omicron BA.2, BA.2.75 and BA.4/5 sublineages carrying those convergent mutations and examined the neutralizing activities of therapeutic neutralizing antibodies (NAbs) against them (Fig. 2a and Extended Data Fig. 2a). In total, pseudoviruses of 50 convergent variants were constructed and tested. COV2-2196 + COV2-2130 (Evusheld)[25] is vulnerable to F486, R346 and K444–G446 mutations, evaded or highly impaired by BJ.1 (R346T), XBB (R346T + V445P + F486S), BA.2.75.2/CA.1/BM.1.1/BM.1.1.1/CH.1.1 (R346T + F486S), CJ.1/XBF (R346T + F486P), BR.2/BR.2.1 (R346T + F486I), BA.4.6.1 (R346T + F486V), BA.5.6.2/BQ.1 (K444T + F486V), BU.1 (K444M + F486V) and BQ.1.1 (R346T + K444T + F486V). LY-CoV1404 (also known as bebtelovimab) remains potent against BF.16 (K444R) and BA.5.5.1 (N450D) and shows reduced potency against BA.5.1.12 (V445A)[26] (Extended Data Fig. 2a). However, LY-CoV1404 was escaped by BJ.1, XBB, BR.1, CH.1.1, BA.4.6.3 and BQ.1.1 while exhibiting strongly reduced activity against BA.2.38.1, BA.5.6.2 and BQ.1 due to K444N/T mutations and the combination of K444M–G446S or V445P–G446S[26]. SA55 + SA58 is a pair of broad NAbs isolated from vaccinated individuals who had SARS that target non-competing conserved epitopes[2,27]. SA58 is weak to G339H and R346T mutations and showed reduced neutralization efficacy against BJ.1/XBB and BA.2.75 sublineages. SA55 is the only NAb demonstrating high potency against all tested Omicron subvariants. Among the tested variants, XBB and BQ.1.1 exhibited the strongest resistance to therapeutic monoclonal Abs (mAbs) and cocktails (Fig. 2a). As the SA55 + SA58 cocktail is still in preclinical development, the efficacy of available antibody drugs, including the BA.2.75/BA.5-effective Evusheld (AstraZeneca) and bebtelovimab, is extensively affected by the emerging subvariants with convergent mutations.

Sufficient ACE2-binding affinity is essential for SARS-CoV-2 transmission. Thus, we examined the relative human ACE2 (hACE2)-binding capability of these variants by evaluating hACE2 inhibitory efficiency against the pseudoviruses. Higher inhibitory efficiency of soluble hACE2 against pseudoviruses indicates higher ACE2-binding capability[28]. Overall, these convergent variants all demonstrate sufficient ACE2-binding efficiency, at least higher than that of D614G, including the most antibody-evasive XBB, BQ.1.1 and CH.1.1 (Fig. 2b and Extended Data Fig. 2b). Specifically, R493Q reversion increases the inhibitory efficiency of hACE2, which is consistent with previous reports[6,20,28].

K417T shows a moderate increase in the inhibitory efficiency of hACE2. By contrast, F486S, K444M and K444N have a clear negative effect on inhibitory efficiency, whereas K444T and F486P do not cause significant impairment of ACE2 binding. These observations are also in line with previous DMS results[29].

Most importantly, we investigated how these variants escape the neutralization of plasma samples from individuals with various immune histories. We recruited cohorts of individuals who received three doses of CoronaVac with or without breakthrough infection by BA.1, BA.2 or BA.5. Convalescent plasma samples were collected on average around 4 weeks after hospital discharge (Supplementary Table 1). Plasma from vaccinated individuals who received CoronaVac was obtained 4 weeks after the third dose. A significant reduction in the 50% neutralization titre ($NT_{50}$) against most tested BA.2, BA.2.75 or BA.5 subvariants was observed, compared with that against corresponding ancestral BA.2, BA.2.75 or BA.5, respectively (Fig. 2c–f and Extended Data Fig. 3a–d).

Specifically, BA.2.3.20 and BA.2.75.2 are significantly more immune-evasive than BA.5 (Fig. 2c–f), explaining their high growth advantage. Nevertheless, multiple convergent variants showed even stronger antibody evasion capability, including BM.1.1.1 (BM.1.1 + F490S), CJ.1/XBF, CA.1 (BA.2.75.2 + L452R + T604I), CA.2 (BA.2.75.2 + S494P), CH.1 (BA.2.75 + R346T + K444T + F486S) and CH.1.1 (CH.1 + L452R) in the BA.2.75 sublineages, and BQ.1.1, BQ.1.1.10 (BQ.1.1 + Y144del) and BA.4.6.3 (BA.4.6 + K444N + N460K + Y144del) in the BA.4/5 sublineages. BN.1 sublineages also caused heavy immune evasion while retaining high hACE2-binding ability. The BJ.1/BM.1.1.1 recombinant strains XBB and XBB.1 (XBB + G252V) are among the most humoral immune-evasive strains tested, comparable with that of CH.1.1, BQ.1.1.10 and BA.4.6.3. BA.5 breakthrough infection yields higher plasma $NT_{50}$ against BA.5 sublineages, including BQ.1.1; however, plasma from individuals who had BA.5 breakthrough infection neutralize poorly against XBB, CH.1.1, BQ.1.1.10 and BA.4.6.3, suggesting that the N-terminal domain (NTD) mutations these variants carry are extremely effective at evading NAbs elicited by BA.5 breakthrough infection (Fig. 2f). Of note, the strongest immune-evasive convergent variants have displayed even lower $NT_{50}$ than SARS-CoV-1, suggesting immense antigenic drift and potential serotype conversion.

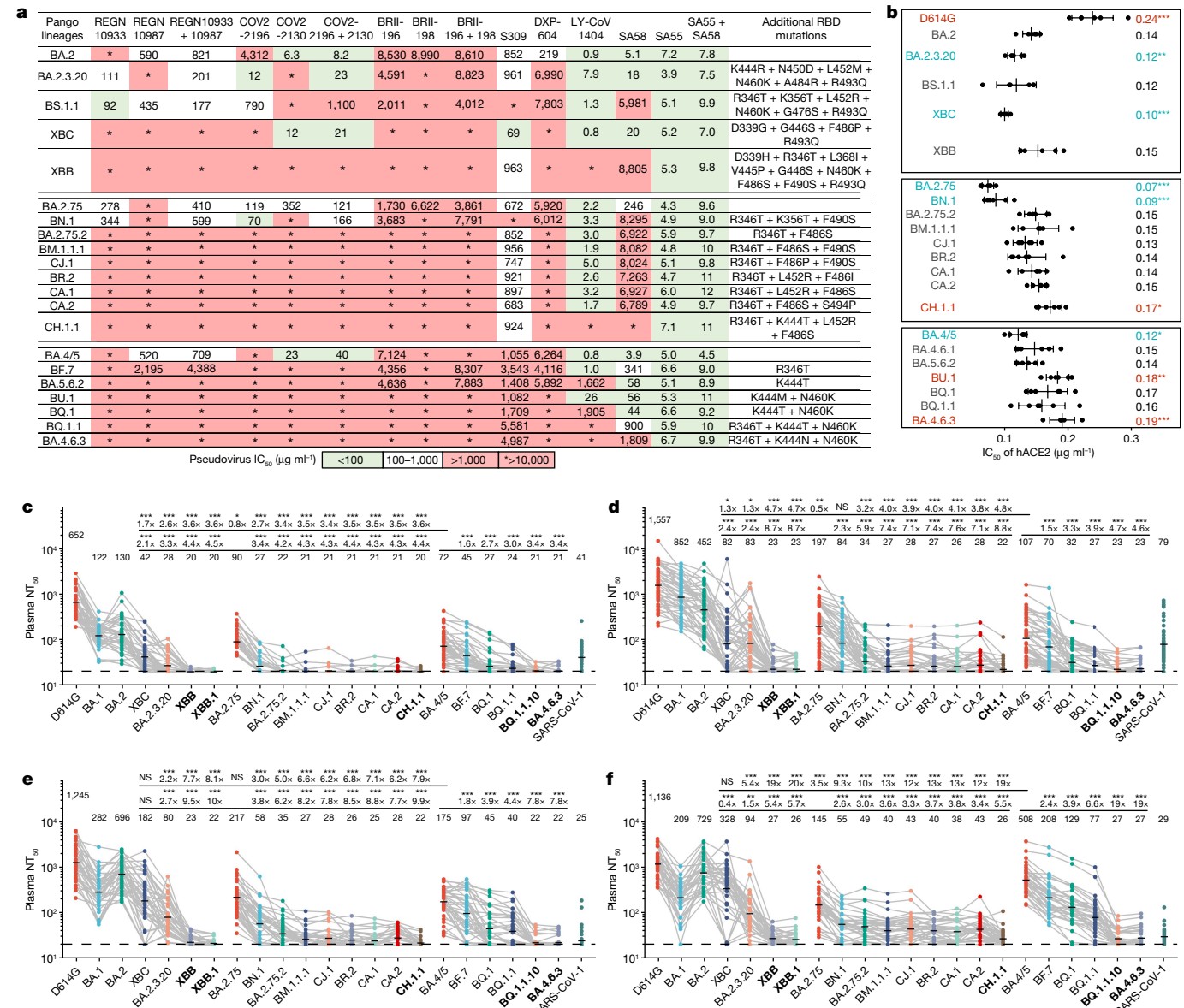

**Fig. 2 | Convergent Omicron subvariants induce NAb evasion. a**, IC$_{50}$ of therapeutic NAbs against VSV-based pseudoviruses with spike glycoproteins of emerging SARS-CoV-2 BA.2, BA.5 or BA.2.75 convergent subvariants. **b**, Relative hACE2-binding capability measured by IC$_{50}$ of hACE2 against pseudoviruses of variants. Error bars indicate mean ± s.d. of $n = 5$ biologically independent replicates. $P$ values were calculated using two-tailed Student's $t$-test. *$P < 0.05$, **$P < 0.01$ and ***$P < 0.001$. There is no label on variants with $P > 0.05$. Variants with significantly stronger binding are coloured blue, whereas those with weaker binding are coloured red. **c–f**, Pseudovirus-neutralizing titres against SARS-CoV-2 D614G and Omicron subvariants of plasma from vaccinated or convalescent individuals of breakthrough infection. Individuals who had received three doses of CoronaVac ($n = 40$) (**c**), individuals

who had been infected with BA.1 after receiving three doses of CoronaVac ($n = 50$) (**d**), individuals who had been infected with BA.2 after receiving three doses of CoronaVac ($n = 39$) (**e**) and individuals who had been infected with BA.5 after receiving three doses of CoronaVac ($n = 36$) (**f**) are shown. The geometric mean titres are labelled. Statistical tests were performed using two-tailed Wilcoxon signed-rank tests of paired samples. *$P < 0.05$, **$P < 0.01$, ***$P < 0.001$ and not significant (NS) $P > 0.05$. NT$_{50}$ against BA.2-derived and BA.2.75-derived subvariants were compared with that against BA.4/5 (the upper line) and BA.2.75 (the lower line); BA.4/5-derived subvariants were only compared with BA.4/5. Dashed lines indicate the limit of detection (LOD, NT$_{50}$ = 20). Strains showing the strongest evasion are in bold. All neutralization assays were conducted in at least two independent experiments.

## RBD convergence due to immune imprinting

It is crucial to investigate the origin of such accelerated RBD convergent evolution. Therefore, we characterized the antibody repertoires induced by Omicron BA.2 and BA.5 breakthrough infection, which is the dominant immune background of current global herd immunity. Following the strategy described in our previous report using pooled peripheral blood mononuclear cells from individuals with BA.1 breakthrough infection[2], we enriched antigen-specific memory B cells by

fluorescence-activated cell sorting (FACS) for individuals who had recovered from BA.2 and BA.5 breakthrough infection (Supplementary Table 1). RBD-binding CD27$^+$/IgM$^-$/IgD$^-$ cells were subjected to single-cell V(D)J sequencing to determine the BCR sequences (Extended Data Fig. 4a,b).

Similar to that reported in BA.1 breakthrough infection, immune imprinting, or 'original antigenic sin', is also observed in BA.2 and BA.5 breakthrough infection[2,30–33]. Post-vaccination infection with BA.2 and BA.5 mainly recalls cross-reactive memory B cells elicited by wild-type

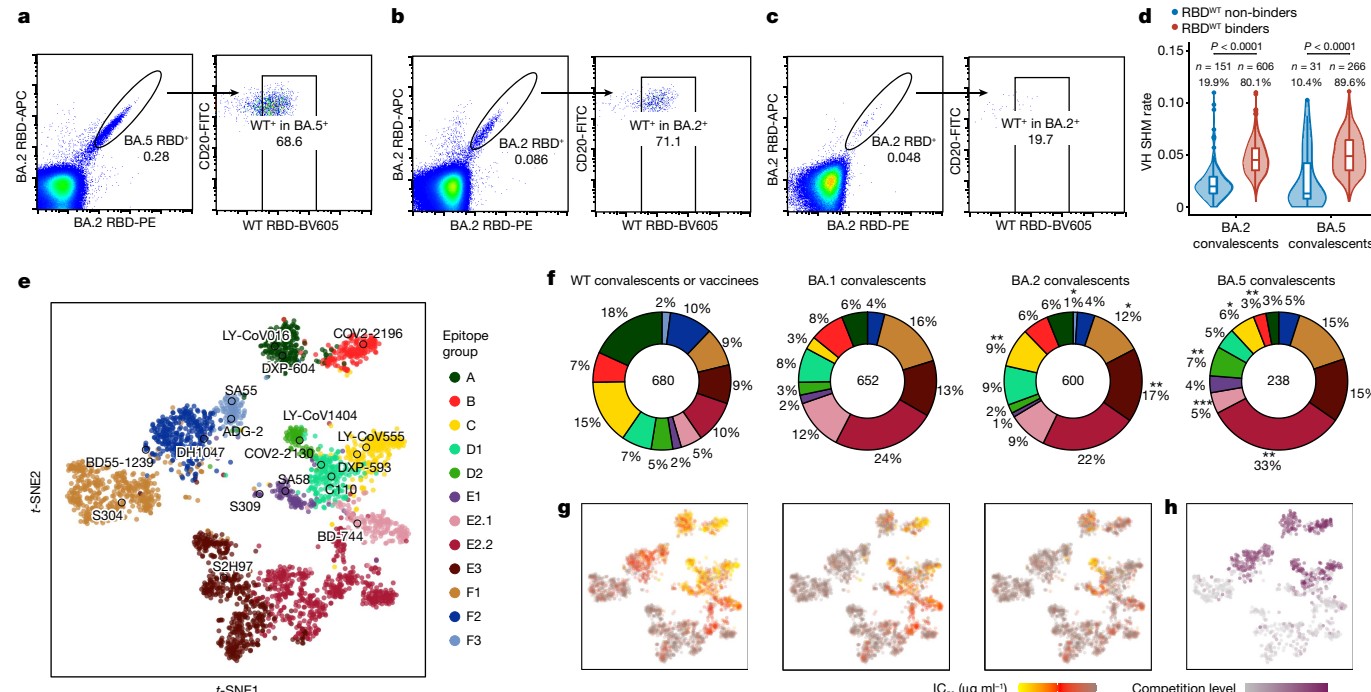

**Fig. 3 | Epitope characterization of mAbs elicited by Omicron breakthrough infections. a–c**, FACS analysis of pooled memory B cells (IgM⁻, IgD⁻/CD27⁺) from Omicron breakthrough-infection convalescent individuals. BA.5 breakthrough infection (**a**), BA.2 breakthrough infection (**b**) and BA.2 convalescent individuals without vaccination (**c**) are shown. APC, allophycocyanine; FITC, fluorescein isothiocyanate; PE, phycoerythrin. **d**, The heavy-chain variable (VH) domain SHM rate of mAbs from individuals with BA.2 (*n* = 757) and BA.5 (*n* = 297) breakthrough infection. Binding specificity was determined by ELISA. Statistical tests were determined using two-tailed Wilcoxon rank-sum tests. Boxes display the 25th percentile, median and 75th percentile, and whiskers indicate median ± 1.5 times the interquartile range. Violin plots show kernel density estimation curves of the distribution. Numbers and ratios of samples in each group are labelled above the violin plots. **e**, *t*-SNE and clustering of SARS-CoV-2 WT RBD-binding antibodies based on DMS profiles of 3,051 antibodies. **f**, Epitope distribution of mAbs from convalescent individuals after WT infection or post-vaccination BA.1, BA.2 or BA.5 infection. Numbers in the centre circles indicate total numbers of mAbs. Colours for epitope groups in **e** also refer to **f**. Two-tailed binomial tests were used to compare the proportion of each epitope group from BA.2 and BA.5 convalescent individuals with that from BA.1. *\*P* < 0.05, \*\**P* < 0.01, \*\*\**P* < 0.001 and no label for *P* > 0.05. **g**, Projection of the neutralizing activity of mAbs against SARS-CoV-2 D614G (left; *n* = 3,046), BA.2.75 (middle; *n* = 3,046) and BA.4/5 (right; *n* = 3,046). **h**, Projection of the ACE2 competition level of mAbs determined by competition ELISA (*n* = 1,317). All neutralization assays and ELISA were conducted in at least two independent experiments.

(WT)-based vaccine, but rarely produces BA.2/BA.5-specific B cells, similar to BA.1 breakthrough infection (Fig. 3a,b). This is in marked contrast to Omicron infection without previous vaccination (Fig. 3c and Extended Data Fig. 4c). The RBD-targeting antibody sequences determined by single-cell V(D)J sequencing are then expressed in vitro as human IgG1 mAbs. As expected, only a small proportion of the expressed mAbs specifically bind to the BA.2/BA.5 RBD and are not cross-reactive to the WT RBD, determined by enzyme-linked immuno-sorbent assay (ELISA) and concordant with the FACS results (Fig. 3d). Cross-reactive mAbs exhibit significantly higher somatic hypermutation (SHM), indicating that these antibodies are more affinity-matured and are indeed most likely recalled from previous vaccination-induced memory (Fig. 3d).

Next, we determined the escape mutation profiles of these antibodies by high-throughput DMS and measured their neutralizing activities against SARS-CoV-2 D614G, BA.2, BA.5, BA.2.75, BQ.1.1 and XBB (Fig. 3e,g and Extended Data Fig. 5a,b). Previously, we reported the DMS profiles and the epitope distribution of antibodies isolated from WT vaccinated or infected individuals, SARS-CoV-2-vaccinated individuals who recovered from SARS, and people who had BA.1 infection, which could be classified into 12 epitope groups². Among them, mAbs in groups A, B, C, D1, D2, F2 and F3 compete with ACE2 and exhibit neutralizing activity (Fig. 3h and Extended Data Figs. 6a–d and 7a–d); whereas mAbs in groups E1, E2.1, E2.2, E3 and F1 do not compete with ACE2 (Fig. 3h and Extended Data Fig. 8a–c). Antibodies in groups E2.2, E3, and F1 exhibit

low or no neutralizing capability (Extended Data Figs. 5b and 8d). To integrate the previous dataset with DMS profiles of the new mAbs isolated from BA.2 and BA.5 convalescent individuals, we co-embedded all antibodies using multidimensional scaling based on their DMS profiles, followed by *t*-distributed stochastic neighbour embedding (*t*-SNE) for visualization, and used *k*-nearest neighbours-based clas-sification to determine the epitope groups of new mAbs (Fig. 3e). This results in a dataset containing the DMS profiles of 3,051 SARS-CoV-2 WT RBD-targeting mAbs in total (Supplementary Table 2). The epitope distribution of mAbs from BA.2 breakthrough infection is generally similar to those elicited by BA.1, except for the increased proportion of mAbs in group C (Fig. 3f). However, BA.5-elicited mAbs showed a more distinct distribution than BA.1, with a significantly increased proportion of mAbs in groups D2 and E2.2, and decreased ratio of antibodies in groups B and E2.1. The main reason is that the F486 and L452 mutations carried by BA.5 make these cross-reactive memory B cells unable to be activated and recalled (Fig. 3f and Extended Data Figs. 6b, 7a and 8a). Antibody repertoires induced by all Omicron breakthrough infections are distinct from those stimulated by WT infection or vaccination. Compared with WT infection or vaccination, BA.1, BA.2 and BA.5 break-through infections mainly elicit mAbs of groups E2.2, E3 and F1, which do not compete with ACE2 and demonstrate weak neutralizing activity, whereas WT-elicited antibodies enrich mAbs of groups A, B and C, which compete with ACE2 and exhibit strong neutralization potency (Fig. 3f–h). The combined proportion of E2.2, E3 and F1 antibodies rose

from 29% in WT convalescent or vaccinated individuals, 53% in BA.1 and 51% in BA.2 convalescent individuals, to 63% in BA.5 convalescent individuals (Fig. 3f). Overall, the proportion and diversity of neutralizing antibody epitopes are reduced in Omicron breakthrough infection, especially in BA.5 breakthrough infection.

To better delineate the effect of immune imprinting and consequent reduction of NAb epitope diversity on the RBD evolutionary pressure, we aggregated the DMS profiles of large collections of mAbs to estimate the effect of mutations on the efficacy of humoral immunity, as inspired by previous works[5] (Supplementary Table 2). It is essential to incorporate the effects of ACE2 binding, RBD expression, neutralizing activity of mAbs and codon usage constraint with the escape profiles to estimate the SARS-CoV-2 evolution trend on the RBD. In brief, each mutation on the RBD would have an effect on each mAb in the set, which is quantified by the escape scores determined by DMS and weighted by its half maximal inhibitory concentration ($IC_{50}$) against the evolving strain. For each residue, only those amino acids that are accessible by one nucleotide mutation are included. The effects on ACE2-binding capability (as measured by pseudovirus inhibitory efficiency) and RBD expression of each mutation are also considered in the analyses, using data determined by DMS in previous reports[4,29,34]. Finally, the estimated relative preference of each mutation is calculated using the sum of weighted escape scores of all mAbs in the specific set.

The reduced NAb epitope diversity caused by imprinted humoral response could be shown by the estimated mutation preference spectrum (Fig. 4a). Diversified escaping-score peaks, which also represent immune pressure, could be observed when using BA.2-elicited antibodies, whereas only two major peaks could be identified, R346T/S and K444E/Q/N/T/M, when using BA.5-elicited antibodies (Fig. 4a). These two hotspots are the most frequently mutated sites in continuously evolving BA.4/BA.5 subvariants, and convergently occurred in multiple lineages (Fig. 1). Similar analyses for WT and BA.1 also demonstrated diversified peaks; thus, the concentrated immune pressure reflects the reduced diversity of NAbs elicited by BA.5 breakthrough infection due to immune imprinting, and these concentrated preferred mutations highly overlapped with convergent hotspots observed in the real world (Extended Data Fig. 9a,b). Together, our results indicate that due to immune imprinting, BA.5 breakthrough infection caused significant reductions in NAb epitope diversity and increased proportion of non-neutralizing mAbs, which in turn focused immune pressure and promoted the convergent RBD evolution.

## Inference of RBD evolution hotspots

Moreover, we wondered whether the real-world evolutionary trends of SARS-CoV-2 RBD could be rationalized and even predicted by aggregating this large DMS dataset containing mAbs from various immune histories. Using the mAbs elicited from WT vaccinated or convalescent individuals weighted by $IC_{50}$ against the D614G strain, we identified mutation hotspots including K417N/T, K444–G446, N450, L452R and especially E484K (Extended Data Fig. 9a). Most of these residues were mutated in previous variants of concern, such as K417N/E484K in Beta, K417T/E484K in Gamma, L452R in Delta and G446S/E484A in Omicron BA.1, confirming our estimation and inference. Evidence of the emergence of BA.2.75 and BA.5 could also be found using mAbs elicited by WT, BA.1 and BA.2 with $IC_{50}$ against BA.2, where peaks on 444–446, 452, 460 and 486 could be identified (Extended Data Fig. 9c). To better investigate the evolution trends of BA.2.75 and BA.5, the two major lineages circulating currently, we then included antibodies elicited by various immune background, including WT/BA.1/BA.2/BA.5 convalescents, which we believe is the best way to represent the current heterogeneous global humoral immunity (Fig. 4b and Extended Data Fig. 9d). For BA.2.75, the most outstanding sites are R346T/S, K356T, N417Y/H/I/T, K444E/Q/N/T/M, V445D/G/A, N450T/D/K/S, L452R, I468N, A484P, F486S/V and F490S/Y. We noticed that these identified residues, even

specific mutations, highly overlapped with recent mutation hotspots of BA.2.75 (Fig. 1). Two exceptions are A484 and I468N. E484 is a featured residue of group C antibodies and could be covered by L452 and F490 (Extended Data Fig. 6c). The I468N mutation is also highly associated with K356 mutations, and its function could be covered by K356T (Extended Data Fig. 8a,b). Owing to stronger antibody evasion, the preference spectrum of BA.5 is much more concentrated than BA.2.75, but the remaining sites are highly overlapped and complementary with BA.2.75. The most striking residues are R346, K444–G446 and N450, followed by K356, N417, L455, N460 and A484. As expected, L452R/F486V does not stand out in the BA.5 preference spectrum, whereas N460K harboured by BA.2.75 appears. These sites and mutations are also popular in emerging BA.4/5 subvariants, proving that our RBD evolution inference system works accurately.

## Evasion mechanism of convergent mutants

It is important to examine where this convergent evolution would lead. On the basis of the observed and predicted convergent hotspots on RBD of BA.2.75 and BA.5, we wondered whether we could construct the convergent variants in advance and investigate to what extent they will evade the humoral immune response. To do this, we must first evaluate the antibody evasion mechanism and the effect on hACE2-binding capability of the convergent mutations and their combinations. Thus, we selected a panel of 178 NAbs from 8 epitope groups that could potently neutralize BA.2 and determined their neutralizing activity against constructed mutants carrying single or multiple convergent mutations (Fig. 4c and Extended Data Fig. 10a). Most of these sites were selected as we have observed at least five independent emergences in distinct lineages of BA.2 and BA.5 that exhibited a growth advantage. NAbs from F1–F3 epitope groups were not included as they are either completely escaped by BA.2 or too rare in individuals convalescent from Omicron breakthrough infection (Fig. 3f). As expected, R493Q and N417T are not major contributors to antibody evasion, but R493Q significantly benefits ACE2 binding. V445A and K444N caused slightly, and F486S/V caused significantly, reduced ACE2-binding capability, consistent with the measurement of emerging subvariants (Figs. 2a and 4c and Extended Data Fig. 2a). The neutralization of NAbs in each group is generally in line with DMS profiles. Most group A NAbs are sensitive to N460K and L455S, and BA.5 + N460K escapes the majority of NAbs in group A owing to the combination of F486V and N460K (Extended Data Fig. 6a). All NAbs in group B are escaped by F486S/V, and group C NAbs are heavily escaped by F490S and are strongly affected by L452R and F486S/V (Extended Data Fig. 6b,c). A part of group C NAbs is also slightly affected by K444N/T, S494P and N450D. G446S affects a part of the D1/D2 NAbs, as previously reported[20]. D1/D2 NAbs are more susceptible to K444N/T, V445A and N450D, and some D1 NAbs could also be escaped by L452R, F490S and S494P (Extended Data Fig. 7a,b). E1 is mainly affected by R346T, D339H and K356T (Extended Data Fig. 7c). E2.1 and E2.2 exhibit similar properties, evaded by K356T, R346T and L452R (Extended Data Fig. 8a,b). E3 antibodies seem to be not strongly affected by any of the constructed mutants, as expected (Extended Data Fig. 8c), but they generally exhibit very low neutralization (Extended Data Fig. 8d). BA.5 + R346T escapes most antibodies in D1, E1 and E2.1/E2.2, and an additional K444N further escapes most mAbs in D2, demonstrating the feasibility and effectiveness of combining convergent mutations to achieve further evasion. Adding six mutations to BA.5 could achieve the evasion of the vast majority of RBD NAbs, while exhibiting high hACE2-binding capability, despite the reduction caused by K444N/T and F486V. BQ.1.1, XBB and CH.1.1 could also escape the majority of RBD-targeting NAbs. Together, these findings indicate the feasibility of generating a heavy-antibody-escaping mutant with accumulated convergent escape mutations while maintaining sufficient hACE2-binding capability (Fig. 4c and Extended Data Fig. 10a).

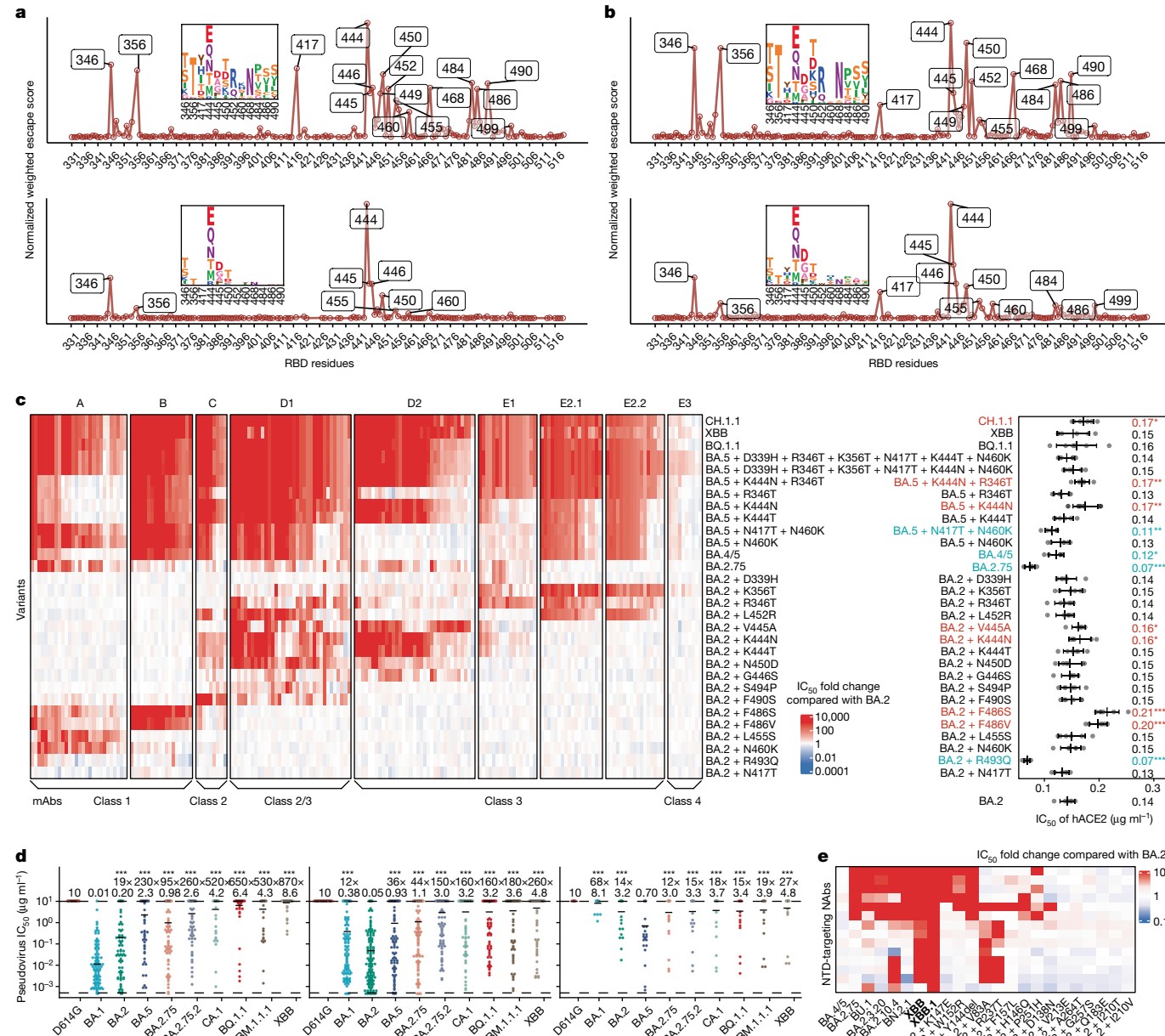

**Fig. 4 | Immune imprinting promotes convergent evolution of NAb-evasive mutations. a,b**, Normalized average escape scores weighted by $IC_{50}$ against BA.2 (top) and BA.5 (bottom) using DMS profiles of NAbs from corresponding convalescent individuals (**a**), and BA.2.75 (top) and BA.5 (bottom) using DMS profiles of all NAbs except those from individuals convalescent from SARS-CoV-1 infection followed by three-dose CoronaVac. (**b**). **c**, $IC_{50}$ of representative potent BA.2-neutralizing antibodies in the epitope group against emerging and constructed Omicron subvariants pseudovirus with escape mutations, in addition to $IC_{50}$ of hACE2 against these variants. The classes of the NAbs as defined in ref. [50] are also annotated below this map. Error bars indicate mean ± s.d. of $n = 5$ biologically independent replicates. $P$ values were calculated using two-tailed Student's $t$-test. *$P < 0.05$, **$P < 0.01$, ***$P < 0.001$ and no label on variants with $P > 0.05$. Variants with significantly stronger binding are coloured blue, whereas those with weaker binding are coloured red. **d**, $IC_{50}$ against featured subvariants of RBD-targeting Omicron-specific NAbs from convalescent individuals with BA.1 (left; $n = 100$), BA.2 (middle; $n = 151$) and BA.5 (right; $n = 31$) breakthrough infection. The geometric mean $IC_{50}$s are labelled, and error bars indicate the geometric standard deviation. Dashed lines indicate the LOD ($IC_{50} = 0.0005$ μg ml⁻¹). $P$ values are calculated using two-tailed Wilcoxon signed-rank tests compared with the corresponding eliciting strain. Antibodies with $IC_{50}$ of more than 10 μg ml⁻¹ against the eliciting strain were excluded from the calculation of $P$ values and fold changes. ***$P < 0.001$ and $P > 0.05$ (NS). **e**, $IC_{50}$ of NTD-targeting NAbs against emerging Omicron subvariants and BA.2 mutants with a single NTD substitution. All neutralization assays were conducted in at least two independent experiments.

Although the proportion of Omicron-specific mAbs is low due to immune imprinting, it is still necessary to evaluate their neutralization potency and breadth, especially against the convergent mutants. We tested the neutralizing activity of a panel of Omicron-specific RBD-targeting mAbs against D614G, BA.1, BA.2, BA.5, BA.2.75, BA.2.75.2, BR.1, BR.2, CA.1, BQ.1.1 and XBB. These mAbs were isolated from convalescent plasma 1 month after Omicron breakthrough infection (Fig. 4d). They could bind to the RBD of the corresponding exposed Omicron variant but did not cross-bind WT RBD, as confirmed by ELISA. We found that these mAbs could effectively neutralize against the exposed strain, as expected, but exhibited poor neutralizing breadth, which means their potency would be largely impaired by other Omicron

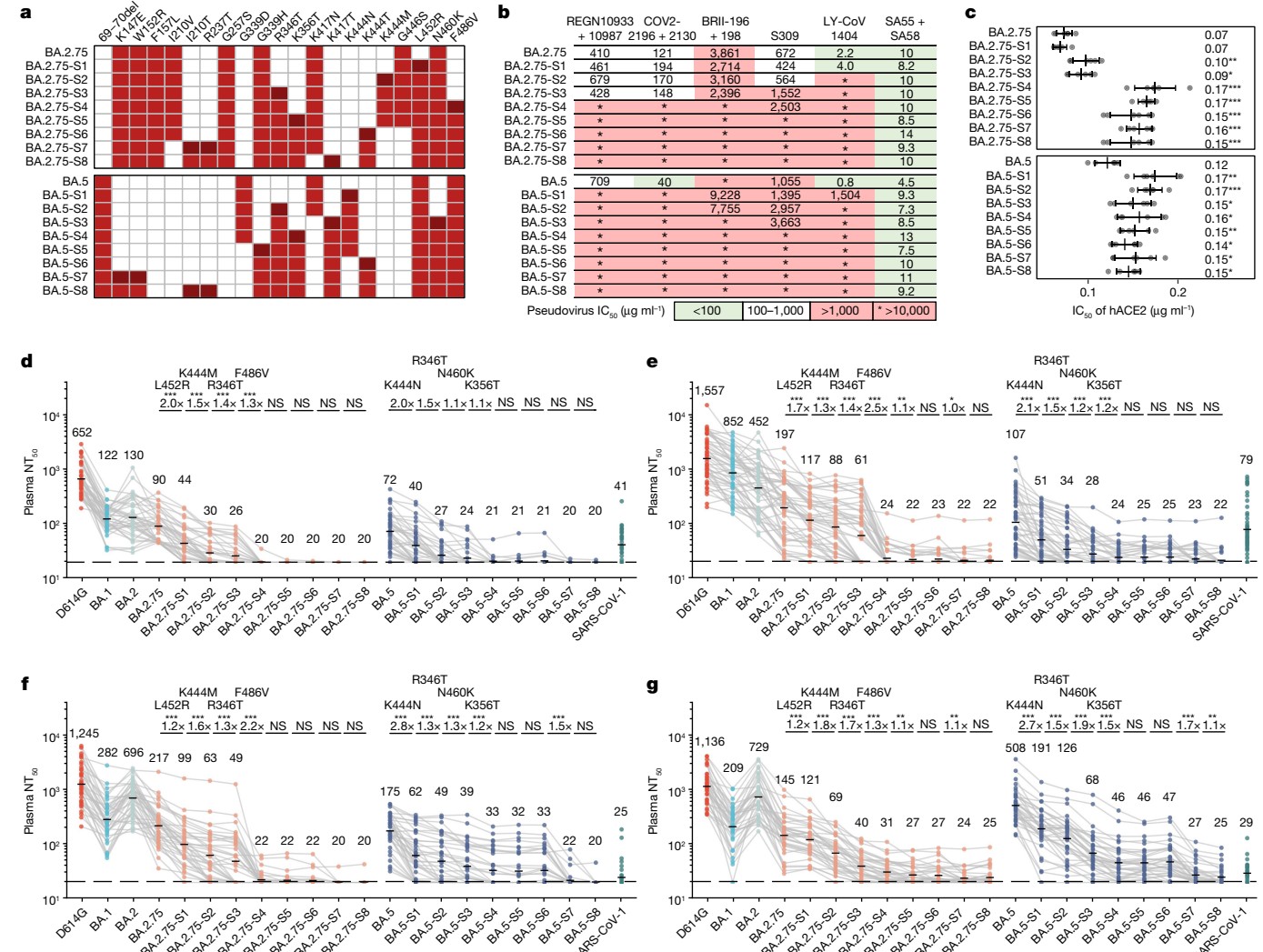

**Fig. 5 | Accumulation of convergent escape mutations leads to complete loss of plasma neutralization. a**, Mutations of multiple designed mutants with key convergent escape mutations based on BA.2.75 and BA.5. Mutations in dark red indicate the additional mutation compared with the former mutant. **b**, IC$_{50}$ of therapeutic mAbs and cocktails against pseudoviruses of designed mutants. **c**, IC$_{50}$ of hACE2 against the designed mutants. Error bars indicate mean ± s.d. of $n = 5$ biologically independent replicates. $P$ values were calculated using two-tailed Student's $t$-test, compared with BA.2.75 and BA.5 for BA.2.75-derived and BA.5-derived mutants, respectively. *$P < 0.05$, **$P < 0.01$, ***$P < 0.001$ and no label on variants with $P > 0.05$. **d–g**, Pseudovirus neutralizing titres against SARS-CoV-2 D614G, Omicron subvariants and designed mutants of plasma from vaccinated or convalescent individuals with breakthrough infection. Individuals who received three doses of CoronaVac ($n = 40$) (**d**), convalescent individuals infected with BA.1 after receiving three doses of CoronaVac ($n = 50$) (**e**), convalescent individuals infected with BA.2 after receiving three doses of CoronaVac ($n = 39$) (**f**), and convalescent individuals infected with BA.5 after receiving three doses of CoronaVac ($n = 36$) (**g**) are shown. Key additional mutations from each designed mutant are annotated above the points. Dashed lines indicate the limit of detection (LOD, NT$_{50} = 20$). The geometric mean titres are labelled. $P$ values are determined using two-tailed Wilcoxon signed-rank tests of paired samples. *$P < 0.05$, **$P < 0.01$, ***$P < 0.001$ and $P > 0.05$ (NS). All neutralization assays were conducted in at least two independent experiments.

subvariants, consistent with our previous discovery[2]. Of note, BQ.1.1 and XBB could escape most of these Omicron-specific NAbs. Thus, these Omicron-specific antibodies would not effectively expand the breadth of the neutralizing antibody repertoire of Omicron breakthrough infection when facing convergent variants. Further affinity maturation may improve the breadth, but additional experiments are needed.

We then evaluated the potency of NTD-targeting NAbs against BA.2, BA.4/5, BA.2.75 and their sublineages and constructed mutants with selected NTD mutations using a panel of 14 NTD-targeting NAbs, as it is reported that NTD-targeting antibodies are abundant in plasma from BA.2 breakthrough infection and contribute to cross-reactivity[35]. Most selected mutations are from recently designated Omicron subvariants, except for R237T, which was near V83A, designed to escape mAbs targeting an epitope recently reported[20]. None of the NTD-targeting

NAbs exhibit strong neutralizing potency, and the IC$_{50}$ values are all over 0.2 μg ml$^{-1}$ (refs. [36,37]) (Fig. 4e and Extended Data Fig. 10b). We found that the tested BA.2-effective NTD-targeting NAbs could be separated into two clusters, named group-α and group-δ in our previous report[20] (Extended Data Fig. 10c). NAbs in group-α target the well-known antigenic supersite on NTD[38], which is sensitive to K147E and W152R in BA.2.75* and Y144del in BJ.1/XBB; whereas group-δ is affected by V83A (XBB) and R237T. The other three NTD mutations in BA.2.75, F157L, I210V and G257S did not obviously affect the tested mAbs, consistent with previous sera neutralization data[28]. Two of the NTD mutations in BJ.1 or XBB, Y144del and V83A, each escapes a cluster of them and together would enable XBB to exhibit extremely strong capability of escaping NTD-targeting NAbs. Of note, XBB.1 escaped all NTD-targeting NAbs tested here.

## Simulating convergent variant evolution

On the basis of the above results, we designed multiple VSV-based pseudoviruses that gradually gain convergent mutations that could induce RBD-targeting or NTD-targeting NAb resistance (Fig. 5a). The constructed final mutant contains 11 additional mutations on the NTD and RBD compared with BA.5, or 9 mutations compared with BA.2.75. The neutralizing activities of Omicron-effective NAb drugs were first evaluated. As expected, the majority of existing effective NAb drugs, except SA55, are escaped by these mutants (Fig. 5b). Similarly, we also determined the ACE2-binding capability of these mutants by neutralization assays using hACE2 (Fig. 5c). Although some of the designed pseudoviruses, especially those with K444N and F486V, exhibit reduced activity to hACE2 compared with the original BA.2.75 or BA.5 variants, their binding affinities are still higher than that of D614G (Fig. 2b). Our designed pseudoviruses could largely evade the plasma of vaccinated individuals and convalescent individuals after BA.1, BA.2 and even BA.5 breakthrough infection (Fig. 5d–g). Among the derivative mutants of BA.2.75, L452R, K444M, R346T and F486V contribute mainly to the significant reduction in neutralization (Fig. 5d–g). Adding more NTD mutations does not contribute to stronger evasion in BA.2.75-based mutants, but we observed a significant reduction in $NT_{50}$ of BA.2/BA.5 convalescent individuals against BA.5-based mutants with K147E + W152R, suggesting that BA.2/BA.5 convalescent plasma contains a large proportion of NTD-targeting antibodies[35]. As the NTD of BA.1 differs from that of BA.2 and BA.5, we did not observe significant effects of NTD mutations on the efficacy of BA.1 convalescent plasma. Plasma neutralization titres of most vaccinated and convalescent individuals decreased to the lower detection limit against BA.2.75 with five extra RBD mutations: L452R, K444M, R346T, F486V and K356T. The same applies to vaccinated individuals or BA.1 convalescents against BA.5 with four extra RBD mutations: K444N, R346T, N460K and K356T. The plasma from BA.2/BA.5 convalescents can tolerate more mutations based on BA.5, and extra NTD mutations such as K147E and W152R are needed to completely eliminate their neutralization. Together, we demonstrate that as few as five additional mutations on BA.5 or BA.2.75 could completely evade most plasma samples, including those from BA.5 breakthrough infection, while maintaining high hACE2-binding capability. Similar efforts have been made in a recent report despite different construction strategies[39]. The constructed evasive mutants, such as BA.2.75-S5/6/7/8 and BA.5-S7/8, could serve to examine the effectiveness of broad-spectrum vaccines and NAbs in advance.

## Discussion

Convergent evolution is common in the biological world, given that one mutation can exhibit strong advantage in particular functions and prevail in multiple lineages. This phenomenon has also been observed in other highly mutated RNA viruses, such as HIV and influenza viruses[40,41]. Previously, N501Y was considered as a convergent mutation that appeared in almost all SARS-CoV-2 variants, which was demonstrated to enhance ACE2-binding affinity[42]. K417 and E484, whose mutations were demonstrated to escape a large number of NAbs, have also exhibited some kind of convergence patterns[43]. However, these previous observations were not outstanding and rapid as recent emergence of convergent mutations on RBD during the global BA.4/5 wave, when several convergent mutations appeared in dozens of sublineages independently, exhibiting growth advantages compared with BA.5. In this work, we showed that due to immune imprinting, our humoral immune repertoire is not effectively diversified by infection with new Omicron variants. The immune pressure on the RBD becomes increasingly concentrated and promotes convergent evolution, explaining the observed sudden acceleration of SARS-CoV-2 RBD evolution and the convergence pattern.

Although this study only examines inactivated vaccines, immune imprinting is also observed in those receiving mRNA vaccines[44,45]. In fact, mRNA-vaccinated individuals displayed an even higher proportion of cross-reactive memory B cells, probably because the overall humoral immune response induced by mRNA vaccines is stronger than that induced by inactivated vaccines[45]. In addition, recent studies on mRNA-vaccinated individuals who receive a BA.5 booster or BA.5 breakthrough infection displayed a similar neutralization reduction trend against BA.2.75.2, BQ.1 and BQ.1.1, suggesting high consistency of neutralization data among vaccine types[46,47].

As the antibodies undergo affinity maturation, their SHM rate would increase[45]. This may lead to a higher proportion of variant-specific antibodies, enhanced binding affinity and increased neutralization breadth, which could potentially resist the convergent mutations carried by variants such as XBB and BQ.1.1 (ref. [48]). However, the effect of affinity maturation may be counteracted by waning immunity[45,49]. The affinity-matured memory B cells would require a second booster or reinfection to be effectively deployed.

We also observed that plasma from individuals with BA.5 breakthrough infection exhibited higher neutralization against BA.5-derived variants such as BQ.1 and BQ.1.1, suggesting that BA.5 boosters and infections are beneficial to protection against convergent variants of BA.5 sublineages. However, this may be mainly driven by the enrichment of NTD-targeting antibodies after BA.5 breakthrough infection, which was also reported in BA.2 convalescent individuals[35]. Specific evasive mutations on the NTD, such as Y144del in XBB and BQ.1.1.10, and mutations of many BA.2.75 sublineages, would cause severe reduction in BA.5 breakthrough infection plasma neutralization titres. Therefore, the effectiveness of BA.5-based boosters against the convergent mutants carrying critical NTD mutations should be closely monitored.

Of note, the antibody evasion capability of many variants, such as BQ.1.1, CA.1, BQ.1.18, XBB and CH.1.1, have already reached or even exceeded SARS-CoV-1, indicating extensive antigenic drift (Fig. 5d–g). Indeed, by constructing an antigenic map of the tested SARS-CoV-2/SARS-CoV-1 variants using the plasma $NT_{50}$ data, we found that the antigenicity distances of SARS-CoV-2 ancestral strain to CA.1, CH.1.1, XBB and BQ.1.1 are already comparable with that of SARS-CoV-1 (Extended Data Fig. 11a,b). Given that there are approximately 50 different amino acids between SARS-CoV-1 and SARS-CoV-2 RBD, but only 21 mutations on the BQ.1.1 RBD compared with the ancestral strain, these results indicate that the global pandemic indeed has greatly promoted the efficiency of the virus to evolve immune escape mutations.

Finally, our prediction demonstrated a remarkable consistency with real-world observations. Some variants close to the predicted and constructed variants have already emerged as we performed the experiments, validating our prediction model. For example, BA.4.6.3 and BQ.1.1 are highly similar to BA.5-S3, and CH.1.1 to BA.2.75-S4/S6 (Fig. 4c). The whole pipeline for constructing pseudoviruses carrying predicted mutations could be safely conducted in biosafety level 2 laboratories, and does not involve any infectious pandemic virus. If we had this prediction model at the beginning of the pandemic, the development of NAb drugs and vaccines might not be so frustrated against the continuously emerging SARS-CoV-2 variants. Broad-spectrum SARS-CoV-2 vaccines and NAb drug development should be of high priority, and the DMS-based prediction of RBD mutations demonstrated in this study could provide effective guidance.

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

## Methods

### Sequence analysis of Omicron sublineages

To identify the sites on RBD with convergence patterns, we first gathered a list of designated Pango lineages and only kept the lineages that exhibited growth advantages over their corresponding ancestral Omicron strains (BA.2, BA.2.75 or BA.5). We identified the parent of each strain according to its Pango lineage full name. Only the additional mutations of each strain compared with its parent are counted in the analysis, which means the inherited mutations will not be counted repeatedly. For example, BQ.1 is the parent of BQ.1.1, and BE.1.1.1 is the parent of BQ.1. Therefore, R346T is the only independent mutation carried by BQ.1.1, and N460K is the only independent mutations carried by BQ.1. K444T is only counted in BE.1.1.1 but not repeatedly counted in BQ.1 and BQ.1.1. For recombinants, the mutations carried by the ancestral recombinant were not counted, but their derivatives were included. Finally, for each site on the RBD, we calculated the number of independent occurrences of mutation on the site, that is, the number of strains that carried mutations on the site and exhibited growth advantage. Sites mutated independently in at least five lineages were considered as convergent mutation sites. The list of strains and the growth advantages over BA.2, BA.2.75 or BA.5 were collected from the #24 collection of CoV-Spectrum (https://cov-spectrum.org)[1]. The full names of designated lineages were collected from the GitHub repository (https://github.com/cov-lineages/pango-designation).

To get the dynamic change of the convergent mutations during the pandemic, spike protein sequences were downloaded from the Global Initiative on Sharing Avian Influenza Data (GISAID; released on 27 October 2022)[16]. The sequences were split according to their date of sampling (from January 2021 to October 2022), and locally aligned to the SARS-CoV-2 WT RBD sequence using biopython (Bio.pairwise2.align.localms, version 1.78) with the scores 2, −1, 8 and 8 for matched, mismatched, gap open and gap extension, respectively. Sequences with an alignment score of less than 200 were excluded from the analysis. 'X' in the sequences was also excluded. The frequencies of each of the 20 amino acids on each RBD site were counted.

### Isolation of peripheral blood mononuclear cells and plasma

Samples from vaccinated individuals and people who had BA.1, BA.2 or BA.5 infection were obtained under study protocols approved by Beijing Ditan Hospital, Capital Medical University (Ethics committee archiving no. LL-2021-024-02) and the Tianjin Municipal Health Commission, and the Ethics Committee of Tianjin First Central Hospital (Ethics committee archiving no. 2022N045KY). All donors provided written informed consent for the collection of information, the use of blood and blood components, and the publication of data generated from this study. Whole-blood samples were diluted 1:1 with PBS + 2% FBS (Gibco) and subjected to Ficoll (Cytiva) gradient centrifugation. Plasma was collected from the upper layer. Cells were collected at the interface and further prepared by centrifugation, red blood cell lysis (Invitrogen eBioscience) and washing steps. The date of vaccination, hospitalization and sampling can be found in Supplementary Table 1.

### BCR sequencing, analysis and antibody production

CD19+ B cells were isolated from peripheral blood mononuclear cells with EasySep Human CD19 Positive Selection Kit II (17854, STEMCELL). Every 10^6 B cells in 100 μl solution were stained with 3 μl FITC anti-human CD20 antibody (clone 2H7, 302304, BioLegend), 3.5 μl Brilliant Violet 421 anti-human CD27 antibody (clone O323, 302824, BioLegend), 2 μl PE/cyanine-7 anti-human IgM antibody (clone MHM-88, 314532, BioLegend), 2 μl PE/cyanine-7 anti-human IgD antibody (clone IA6-2, 348210, BioLegend), 0.013 μg biotinylated SARS-CoV-2 BA.2 RBD protein (customized from Sino Biological) or 0.013 μg biotinylated SARS-CoV-2 BA.5 RBD protein (customized from Sino Biological) conjugated with PE-streptavidin (405204, BioLegend) and APC-streptavidin

(405207, BioLegend), and 0.013 μg SARS-CoV-2 WT biotinylated RBD protein (40592-V27H-B, Sino Biological) conjugated with Brilliant Violet 605 streptavidin (405229, BioLegend). Cells were also labelled with biotinylated RBD conjugated to DNA-oligo-streptavidin. Omicron RBD (BA.2 or BA.5) were labelled with TotalSeq-C0971 streptavidin (405271, BioLegend) and TotalSeq-C0972 streptavidin (405273, BioLegend); WT RBD were labelled with TotalSeq-C0973 streptavidin (405275, BioLegend) and TotalSeq-C0974 streptavidin (405277, BioLegend). Cells were washed twice after 30 min of incubation on ice. 7-AAD (00-6993-50, Invitrogen) was used to label dead cells. 7-AAD−CD20+CD27+IgM− IgD− SARS-CoV-2 BA.2 RBD+ or BA.5 RBD+ cells were sorted with a MoFlo Astrios EQ Cell Sorter. FACS data were collected by Summit 6.0 (Beckman Coulter). FACS data were analysed using FlowJo v10.8 (BD Biosciences).

Sorted B cells were resuspended in the appropriate volume and then processed with Chromium Next GEM Single Cell V(D)J Reagent Kits v1.1 following the manufacturer's user guide (CG000208, 10X Genomics). Gel beads-in-emulsion (GEMs) were obtained with a 10X Chromium controller. GEMs were subjected to reverse transcription and purification. Reverse transcription products were subject to pre-amplification and purification with the SPRIselect Reagent Kit (B23318, Beckman Coulter). BCR sequences (paired V(D)J) were enriched with 10X BCR primers. After library preparation, libraries were sequenced with the Illumina sequencing platform.

10X Genomics V(D)J sequencing data were assembled as BCR contigs and aligned using the Cell Ranger (v6.1.1) pipeline according to the GRCh38 BCR reference. Only the productive contigs and the B cells with one heavy chain and one light chain were kept for quality control. The germline V(D)J gene identification and annotation were performed by IgBlast (v1.17.1)[51]. Somatic hypermutation sites in the antibody variable domain were detected using the Change-O toolkit (v1.2.0)[52].

Antibody heavy-chain and light-chain genes were optimized for human cell expression and synthesized by GenScript. VH and VL were inserted separately into plasmids (pCMV3-CH, pCMV3-CL or pCMV3-CK) through infusion (C112, Vazyme). Plasmids encoding the heavy chain and light chain of antibodies were co-transfected by polyethylenimine transfection to Expi293F cells (A14527, Thermo Fisher). Cells were cultured at 36.5 °C in 5% $CO_2$ at 175 r.p.m. for 6–10 days. Supernatants containing mAbs were collected and the supernatants were further purified with protein A magnetic beads (L00695, Genscript).

### High-throughput DMS

The high-throughput DMS platform has been previously described[2,3]. In brief, DMS libraries were constructed by mutagenesis PCR based on the Wuhan-Hu-1 RBD sequence (residues N331–T531; GenBank: MN908947). A unique 26-nucleotide (N26) barcode was appended to each RBD variant in mutant libraries by PCR, and the correspondence between the N26 barcode and mutations in RBD variants was acquired by PacBio sequencing. RBD-mutant libraries were first transformed in the EBY100 strain of *Saccharomyces cerevisiae* and then enriched for properly folded ACE2 binders, which were used for subsequent mutation escape profiling. The above ACE2 binders were grown in SG-CAA medium (2% w/v D-galactose, 0.1% w/v dextrose (D-glucose), 0.67% w/v yeast nitrogen base, 0.5% w/v casamino acids (-ade, -ura and -trp) and 100 mM phosphate buffer, pH 6.0) at room temperature for 16–18 h with agitation. Then, these yeast cells were washed twice and proceeded to three rounds of magnetic beads-based selection. Obtained yeast cells after sequential sorting were recovered overnight in SD-CAA medium (2% w/v dextrose (D-glucose), 0.67% w/v yeast nitrogen base, 0.5% w/v casamino acids (-ade, -ura and -trp) and 70 mM citrate buffer, pH 4.5). Pre-sorting and post-sorting yeast populations were submitted to plasmid extraction by a 96 Well Plate Yeast Plasmid Preps Kit (PE053, Coolaber). N26 barcode sequences were amplified with the extracted plasmid templates, and PCR products were purified and submitted to Illumina Nextseq 550 sequencing.

### Antibody clustering and embedding based on DMS profiles

Data analysis of DMS was performed as described in previous reports[2,3]. In brief, the detected barcode sequences of both the antibody-screened and reference library were aligned to the barcode-variant lookup table generated using dms_variants (v0.8.9). The escape scores of each variant $X$ in the library were defined as $F \times (n_{X,ab} / N_{ab}) / (n_{X,ref} / N_{ref})$, where $F$ is a scale factor to normalize the scores to the 0–1 range, and $n$ and $N$ are the number of detected barcodes for variant $X$ and total barcodes in post-selected (ab) or reference (ref) samples, respectively. The escape scores of each mutation were calculated by fitting an epistasis model as previously described[4,53].

Epitope groups of new antibodies not included in our previous report are determined by the $k$-nearest neighbours-based classification. In brief, site escape scores of each antibody are first normalized and considered as a distribution across RBD residues, and only residues whose standard derivation is among the highest 50% of all residues are retained for further analysis. Then, the dissimilarity or distance of two antibodies is defined by the Jessen–Shannon divergence of the normalized escape scores. Pairwise dissimilarities of all antibodies in the dataset are calculated using the scipy package (scipy.spatial.distance.jensenshannon, v1.7.0). For each antibody, the 15 nearest neighbours whose epitope groups have been determined by unsupervised clustering in our previous paper were identified and simply voted to determine the group of the selected antibody. To project the dataset onto a 2D space for visualization, we performed multidimensional scaling to represent each antibody in a 32-dimensional space, and then $t$-SNE to get the 2D representation, using sklearn.manifold.MDS and sklearn.manifold.TSNE (v0.24.2). Figures were generated by R package ggplot2 (v3.3.3).

### Calculation of the estimated preference of RBD mutations

Four different weights are included in the calculation, including the weight for ACE2-binding affinity, RBD expression, codon constraint and neutralizing activity. The effect on ACE2-binding affinity and RBD expression of each mutation based on WT, BA.1 and BA.2 were obtained from public DMS results. For BA.5 (BA.2 + L452R + F486V + R493Q) and BA.2.75 (BA.2 + D339H + G446S + N460K + R493Q), BA.2 results were used except for these mutated residues, whose scores for each mutant were subtracted by the score for the mutation in BA.5 or BA.2.75. As the reported values are log fold changes, the weight is simply defined by the exponential of reported values, that is, $\exp(S_{bind})$ or $\exp(S_{expr})$, respectively. For codon constraint, the weight is 1.0 for mutants that could be accessed by one nucleotide mutation, and 0.0 for others. We used the following RBD nucleotide sequences for determination of accessible mutants: WT/D614G (Wuhan-Hu-1 reference genome), BA.1 (EPI_ISL_10000028), BA.2 (EPI_ISL_10000005), BA.4/5 (EPI_ISL_11207535) and BA.2.75 (EPI_ISL_13302209). For neutralizing activity, the weight is $-\log_{10}(IC_{50})$. The $IC_{50}$ values ($\mu g\ ml^{-1}$), which are smaller than 0.0005 or larger than 1.0 are considered as 0.0005 or 1.0, respectively. The raw escape scores for each antibody were first normalized by the maximum score among all mutants, and the final weighted score for each antibody and each mutation is the product of the normalized scores and four corresponding weights. The final mutation-specific weighted score is the summation of scores of all antibodies in the designated antibody set and then normalized again to make it a value between 0 and 1. Logo plots for visualization of escape maps were generated by the Python package logomaker (v0.8).

### Pseudovirus neutralization assay

The gene encoding the spike protein (GenBank: MN908947) was mammalian codon-optimized and inserted into the pcDNA3.1 vector. Site-directed mutagenesis PCR was performed as previously described[54]. The sequence of mutants is shown in Supplementary Table 3.

Pseudotyped viruses were generated by transfection of 293T cells (CRL-3216, American Type Culture Collection) with pcDNA3.1-spike with Lipofectamine 3000 (Invitrogen). The cells were subsequently infected with G*ΔG-VSV (Kerafast) that packages expression cassettes for firefly luciferase instead of VSV-G in the VSV genome. The cell supernatants were discarded after 6–8 h of harvest and replaced with complete culture media. The cell was cultured for 1 day, and then the cell supernatant containing pseudotyped virus was harvested, filtered (0.45-μm pore size; Millipore), aliquoted and stored at −80 °C. Viruses of multiple variants were diluted to the same number of copies before use.

mAbs or plasma were serially diluted and incubated with the pseudotyped virus in 96-well plates for 1 h at 37 °C. Trypsin-treated Huh-7 cells (0403, Japanese Collection of Research Bioresources) were added to the plate. The cells were cultured for 20–28 h in 5% $CO_2$ and 37 °C incubators. The supernatants were removed and left 100 μl in each well, and 100 μl luciferase substrate (6066769, PerkinElmer) was added and incubated in the dark for 2 min. The cell lysate was removed and the chemiluminescence signals were collected by PerkinElmer Ensight. Each experiment was repeated at least twice.

Inhibitory efficiencies of hACE2 against the pseudoviruses were determined with the same procedure, using the hACE2-Fc dimer (10108-H02H, Sino Biological), and each experiment was conducted in five biologically independent replicates.

DMEM (high glucose; HyClone) with 100 U ml$^{-1}$ penicillin–streptomycin solution (Gibco), 20 mM HEPES (Gibco) and 10% FBS (Gibco) was used in cell culture. Trypsin-EDTA (0.25%; Gibco) was used to detach cells before seeding to the plate.

### ELISA

WT/BA.2/BA.1 RBD or spikes in PBS was pre-coated onto ELISA plates at 4 °C overnight and were washed and blocked. Purified antibodies (1 μg ml$^{-1}$) were added and incubated at room temperature for 20 min. Peroxidase-conjugated AffiniPure goat anti-human IgG (H+L) (0.25 μg ml$^{-1}$; 109-035-003, JACKSON) was added to plates and incubated at room temperature for 15 min. Tetramethylbenzidine (54827-17-7, Solarbio) was added and incubated for 10 min, and then the reaction was terminated with 2 M $H_2SO_4$. Absorbance was measured at 450 nm using a microplate reader (HH3400, PerkinElmer). H7N9 human IgG1 antibody HG1K (1 μg ml$^{-1}$; HG1K, Sino Biological) was used as negative control.

### Reporting summary

Further information on research design is available in the Nature Portfolio Reporting Summary linked to this article.

## Data availability

Processed mutation escape scores can be downloaded at https://github.com/jianfcpku/convergent_RBD_evolution. Sequences and neutralization of the antibodies are included in Supplementary Table 2. Raw sequencing data of DMS assays are available on China National GeneBank (db.cngb.org) with project accession CNP0003808. We used vdj_GRCh38_alts_ensembl-5.0.0 as the reference for V(D)J alignment, which can be obtained from https://support.10xgenomics.com/single-cell-vdj/software/downloads/latest. We used Protein Data Bank ID 6M0J for the structural model of the SARS-CoV-2 RBD. A list of strains and the growth advantages was collected from the #24 collection from https://cov-spectrum.org. Designated lineages were from https://github.com/cov-lineages/pango-designation.

## Code availability

Custom scripts to analyse the escape mutation profile data are available at https://github.com/jianfcpku/convergent_RBD_evolution.

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

**Acknowledgements** We thank J. Bloom for his gift of the yeast SARS-CoV-2 RBD libraries; all volunteers for providing the blood samples; and all the scientists around the globe for performing SARS-CoV-2 sequencing and surveillance analysis. This project is financially supported by the Ministry of Science and Technology of China and Changping Laboratory (2021A0201 and 2021D0102), and the National Natural Science Foundation of China (32222030).

**Author contributions** Y.C. designed the study. X.S.X. supervised the study. Y.C., F.J., A.Y., Q.G. and X.S.X. wrote the manuscript with input from all authors. A.Y., W.S., R.A., Yao Wang and X.N. performed B cell sorting, single-cell VDJ sequencing and antibody sequence analyses. J.W. (BIOPIC), H.S. and F.J. performed and analysed the DMS data. Y.Y. and Youchun Wang constructed the pseudotyped virus. N.Z., P.W., L.Y., T.X. and F.S. performed the pseudotyped virus neutralization assays. W.S. and Y.C. analysed the neutralization data. X.H., Y.X., X.C., Z.S. and R.J. recruited the SARS-CoV-2 vaccinated and convalescent individuals. J.W. (Changping Laboratory), L.Y. and F.S. performed the antibody expression.

**Competing interests** X.S.X. and Y.C. are founders of Singlomics Biopharmaceuticals. Changping Laboratory is in the process of applying for provisional patents (PCT/CN2021/090146 and PCT/CN2021/080537) covering BD series SARS-CoV-2 monoclonal antibodies, including BD-604 (DXP-604), BD55-5840 (SA58) and BD55-5514 (SA55), that lists X.S.X. and Y.C. as inventors. All other authors declare no competing interests.

**Additional information**
**Correspondence and requests for materials** should be addressed to Yunlong Cao, Youchun Wang or Xiaoliang Sunney Xie.

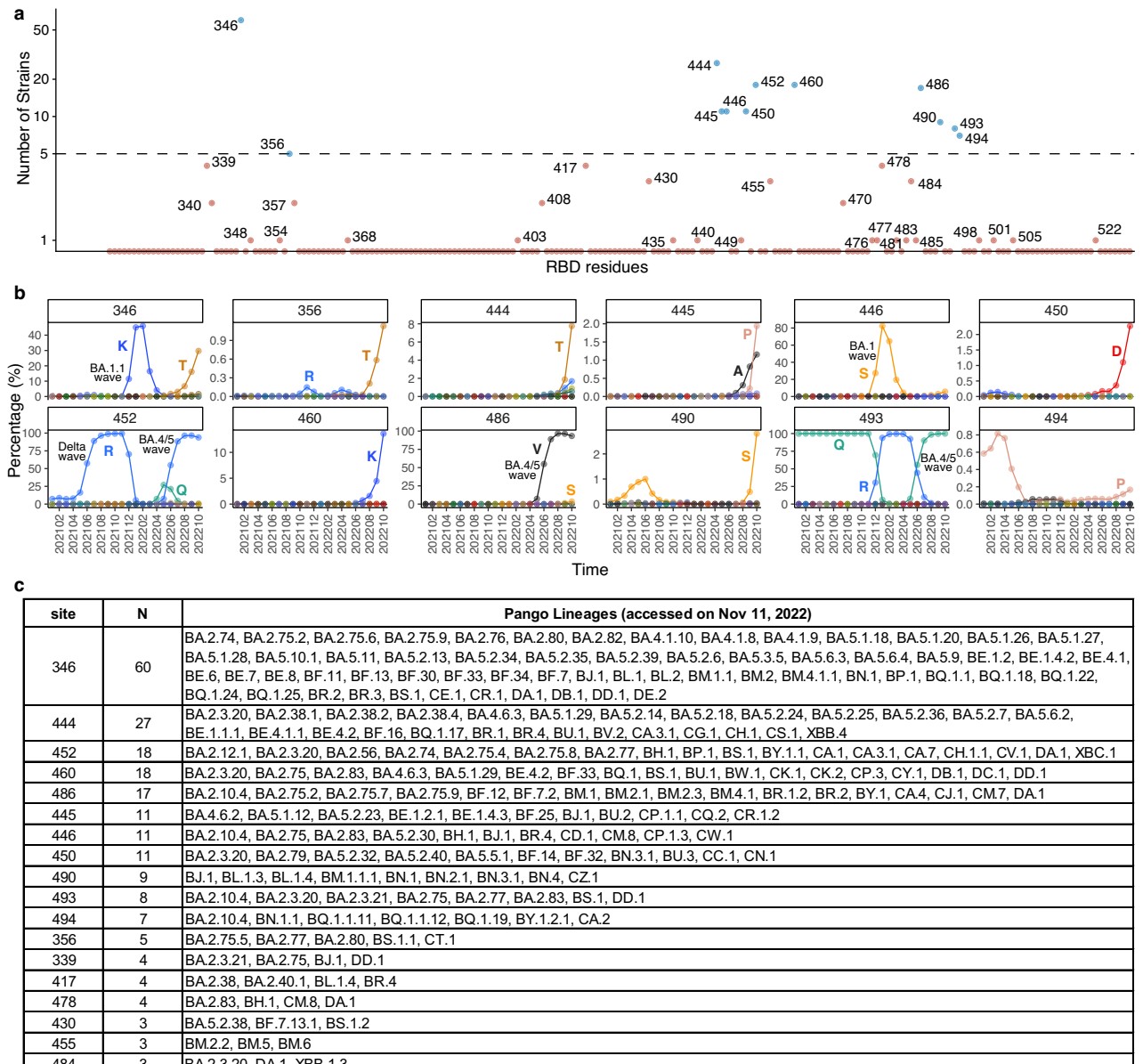

| site | N | Pango Lineages (accessed on Nov 11, 2022) |
|---|---|---|
| 346 | 60 | BA.2.74, BA.2.75.2, BA.2.75.6, BA.2.75.9, BA.2.76, BA.2.80, BA.2.82, BA.4.1.10, BA.4.1.8, BA.4.1.9, BA.5.1.18, BA.5.1.20, BA.5.1.26, BA.5.1.27, BA.5.1.28, BA.5.10.1, BA.5.11, BA.5.2.13, BA.5.2.34, BA.5.2.35, BA.5.2.39, BA.5.2.6, BA.5.3.5, BA.5.6.3, BA.5.6.4, BA.5.9, BE.1.2, BE.1.4.2, BE.4.1, BE.6, BE.7, BE.8, BF.11, BF.13, BF.30, BF.33, BF.34, BF.7, BJ.1, BL.1, BL.2, BM.1.1, BM.2, BM.4.1.1, BN.1, BP.1, BQ.1.1, BQ.1.18, BQ.1.22, BQ.1.24, BQ.1.25, BR.2, BR.3, BS.1, CE.1, CR.1, DA.1, DB.1, DD.1, DE.2 |
| 444 | 27 | BA.2.3.20, BA.2.38.1, BA.2.38.2, BA.2.38.4, BA.4.6.3, BA.5.1.29, BA.5.2.14, BA.5.2.18, BA.5.2.24, BA.5.2.25, BA.5.2.36, BA.5.2.7, BA.5.6.2, BE.1.1.1, BE.4.1.1, BE.4.2, BF.16, BQ.1.17, BR.1, BR.4, BU.1, BV.2, CA.3.1, CG.1, CH.1, CS.1, XBB.4 |
| 452 | 18 | BA.2.12.1, BA.2.3.20, BA.2.56, BA.2.74, BA.2.75.4, BA.2.75.8, BA.2.77, BH.1, BP.1, BS.1, BY.1.1, CA.1, CA.3.1, CA.7, CH.1.1, CV.1, DA.1, XBC.1 |
| 460 | 18 | BA.2.3.20, BA.2.75, BA.2.83, BA.4.6.3, BA.5.1.29, BE.4.2, BF.33, BQ.1, BS.1, BU.1, BW.1, CK.1, CK.2, CP.3, CY.1, DB.1, DC.1, DD.1 |
| 486 | 17 | BA.2.10.4, BA.2.75.2, BA.2.75.7, BA.2.75.9, BF.12, BF.7.2, BM.1, BM.2.1, BM.2.3, BM.4.1, BR.1.2, BR.2, BY.1, CA.4, CJ.1, CM.7, DA.1 |
| 445 | 11 | BA.4.6.2, BA.5.1.12, BA.5.2.23, BE.1.2.1, BE.1.4.3, BF.25, BJ.1, BU.2, CP.1.1, CQ.2, CR.1.2 |
| 446 | 11 | BA.2.10.4, BA.2.75, BA.2.83, BA.5.2.30, BH.1, BJ.1, BR.4, CD.1, CM.8, CP.1.3, CW.1 |
| 450 | 11 | BA.2.3.20, BA.2.79, BA.5.2.32, BA.5.2.40, BA.5.5.1, BF.14, BF.32, BN.3.1, BU.3, CC.1, CN.1 |
| 490 | 9 | BJ.1, BL.1.3, BL.1.4, BM.1.1.1, BN.1, BN.2.1, BN.3.1, BN.4, CZ.1 |
| 493 | 8 | BA.2.10.4, BA.2.3.20, BA.2.3.21, BA.2.75, BA.2.77, BA.2.83, BS.1, DD.1 |
| 494 | 7 | BA.2.10.4, BN.1.1, BQ.1.1.11, BQ.1.1.12, BQ.1.19, BY.1.2.1, CA.2 |
| 356 | 5 | BA.2.75.5, BA.2.77, BA.2.80, BS.1.1, CT.1 |
| 339 | 4 | BA.2.3.21, BA.2.75, BJ.1, DD.1 |
| 417 | 4 | BA.2.38, BA.2.40.1, BL.1.4, BR.4 |
| 478 | 4 | BA.2.83, BH.1, CM.8, DA.1 |
| 430 | 3 | BA.5.2.38, BF.7.13.1, BS.1.2 |
| 455 | 3 | BM.2.2, BM.5, BM.6 |
| 484 | 3 | BA.2.3.20, DA.1, XBB.1.3 |

**Extended Data Fig. 1 | Emergence of convergent mutations on SARS-CoV-2 RBD. a**, Number of independent Omicron sublineages that gained mutations on the corresponding SARS-CoV-2 RBD residue and exhibited growth advantage compared to its ancestral Omicron strain (BA.2, BA.2.75, or BA.5). Residues that were mutated in at least five independent sublineages are considered convergent (dash threshold). Recombinants were not counted, but their derivatives were included in the analysis (see Methods). **b**, Proportions of each convergent mutation in all detected Spike sequences. Spike sequences were from GISAID (Spike protein sequences released on Oct 27, 2022). The percentage of the wild-type residue is not plotted, except for 493Q, considering the prevalence of R493Q reversion. **c**, List of Pango lineages shown in Extended Data Fig. 1a.

**a**

| Pango lineages | REGN 10933 | REGN 10987 | REGN10933 +10987 | COV2 -2196 | COV2 -2130 | COV2- 2196+2130 | BRII- 196 | BRII- 198 | BRII- 196+198 | S309 | DXP- 604 | LY-CoV 1404 | SA58 | SA55 | SA55+ SA58 | Additional RBD mutations |
|---|---|---|---|---|---|---|---|---|---|---|---|---|---|---|---|---|
| BA.2 | * | 590 | 821 | 4312 | 6.3 | 8.2 | 8530 | 8990 | 8610 | 852 | 219 | 0.9 | 5.1 | 7.2 | 7.8 | |
| BA.2.10.4 | * | * | * | * | 289 | 501 | 2109 | 7990 | 3984 | 706 | 6348 | 1.3 | 4.3 | 4.9 | 5.0 | G446S+F486P+R493Q+S494P |
| BA.2.38.1 | * | * | * | 3391 | * | 4736 | 4571 | * | 6212 | 876 | 33 | 1504 | 83 | 11 | 13 | N417T+K444N |
| BA.2.74 | * | 2250 | 3211 | 3214 | * | 4300 | 6780 | * | * | 4943 | 191 | 2.6 | 293 | 4.2 | 6.9 | R346T+L452M |
| BA.2.76 | * | 2336 | 4241 | 3162 | * | 5450 | 6021 | * | * | 6303 | 181 | 1.3 | 376 | 3.8 | 5.2 | R346T |
| BA.2.77 | * | 657 | 584 | 4745 | 24 | 45 | 7209 | * | * | * | 259 | 0.9 | * | 4.6 | 6.6 | K356R+L452R+E340K |
| BA.2.79 | * | 3051 | 7703 | 3587 | 3633 | 3019 | 3797 | * | 7767 | 1023 | 178 | 14 | 6.1 | 7.2 | 4.9 | N450D |
| BA.2.80 | * | 2096 | 2201 | 4022 | * | 4133 | 5096 | * | 7202 | * | 206 | 1.0 | * | 6.2 | 10 | R346T+K356R+E340K |
| BJ.1 | * | * | * | 3076 | * | 5985 | 7609 | * | * | 709 | 166 | * | 8163 | 3.7 | 8.6 | D339H+R346T+L368I+ V445P+G446S+V483A+ F490V |
| BA.2.75 | 278 | * | 410 | 119 | 352 | 121 | 1730 | 6622 | 3861 | 672 | 5920 | 2.2 | 246 | 4.3 | 9.6 | |
| BA.2.75.4 | 249 | * | 461 | * | 80 | 194 | 1448 | * | 2714 | 424 | 6553 | 4.0 | 254 | 4.0 | 8.2 | L452R |
| BA.2.75.5 | 245 | * | 410 | 101 | 550 | 188 | 1590 | * | 3699 | 4780 | 6691 | 2.2 | 3305 | 4.8 | 9.1 | K356T |
| BN.2.1 | 390 | * | 701 | 59 | 303 | 109 | 4101 | * | 8444 | 6979 | 8901 | 1.7 | 4960 | 5.7 | 9.4 | K356T+F490S |
| BL.1 | 260 | * | 511 | 93 | * | 174 | 1251 | * | 3075 | 508 | 7193 | 2.8 | 7975 | 6.3 | 10 | R346T |
| BR.1 | 319 | * | 679 | 117 | * | 170 | 1992 | * | 3160 | 564 | 6689 | * | 1616 | 5.9 | 9.7 | L452R+K444M |
| BM.1 | * | * | * | * | 301 | 563 | * | * | * | 796 | * | 1.9 | 330 | 7.0 | 10.0 | F486S |
| BM.1.1 | * | * | * | * | * | * | * | * | * | 879 | * | 2.3 | 8823 | 5.2 | 8.9 | R346T+F486S |
| CH.1 | * | * | * | * | * | * | * | * | * | 990 | * | * | * | 6.0 | 10 | R346T+K444T+F486S |
| BA.4/5 | * | 520 | 709 | * | 23 | 40 | 7124 | * | * | 1055 | 6264 | 0.8 | 3.9 | 5.0 | 4.5 | |
| BF.16 | * | 883 | 1863 | * | 48 | 79 | 4715 | * | 5507 | 1575 | 7505 | 2.7 | 6.7 | 3.3 | 6.0 | K444R |
| BA.5.2.7 | * | * | * | * | * | * | 3701 | * | 6502 | 1419 | 6263 | 25 | 63 | 4.3 | 9.6 | K444M |
| BA.5.1.12 | * | * | * | * | 1843 | 4818 | 3505 | * | 4849 | 1752 | 8268 | 98 | 3.6 | 3.4 | 4.6 | V445A |
| BA.5.5.1 | * | 1936 | 2963 | * | * | * | 5023 | * | 8236 | 1293 | 6807 | 5.1 | 17 | 6.0 | 8.0 | N450D |
| BA.4.7 | * | 1362 | 2420 | * | * | * | 3395 | * | 5209 | 2899 | 5189 | 1.4 | 598 | 3.8 | 6.8 | R346S |
| BA.5.9 | * | 2701 | 3498 | * | * | * | 4023 | * | 7695 | 4780 | 8360 | 1.3 | 404 | 4.8 | 8.7 | R346I |
| BA.5.6.4 | * | 1096 | 2291 | * | * | * | 3127 | * | 6669 | 5598 | 7988 | 0.9 | * | 3.8 | 7.8 | R346E |
| BE.4.1.1 | * | 5133 | * | * | * | * | 3192 | * | 6042 | 2764 | 6896 | 8.4 | 798 | 4.8 | 7.9 | R346T+K444R |

Pseudovirus IC50 (ng/mL): <100 | 100~1,000 | >1,000 | * >10,000

**b** IC50 of hACE2 (µg/mL)

| Variant | IC50 of hACE2 (µg/mL) |
|---|---|
| BA.2 | 0.14 |
| BA.2.10.4 | 0.11** |
| BA.2.38.1 | 0.16* |
| BA.2.74 | 0.12* |
| BA.2.76 | 0.13 |
| BA.2.77 | 0.15 |
| BA.2.79 | 0.15 |
| BA.2.80 | 0.13 |
| BJ.1 | 0.16 |
| BA.2.75 | 0.07*** |
| BA.2.75.4 | 0.07*** |
| BA.2.75.5 | 0.08*** |
| BN.2.1 | 0.08*** |
| BL.1 | 0.09*** |
| BR.1 | 0.10*** |
| BM.1 | 0.16 |
| BM.1.1 | 0.16 |
| CH.1 | 0.17** |
| BA.4/5 | 0.12* |
| BF.16 | 0.15 |
| BA.5.2.7 | 0.17* |
| BA.5.1.12 | 0.17 |
| BA.5.5.1 | 0.15 |
| BA.4.7 | 0.15 |
| BA.5.9 | 0.14 |
| BA.5.6.4 | 0.13 |
| BE.4.1.1 | 0.16 |

**Extended Data Fig. 2 | Antibody drug evasion and hACE2 binding capability of convergent Omicron variants. a**, IC50 of therapeutic NAbs against pseudoviruses of additional emerging SARS-CoV-2 Omicron subvariants. **b**, Relative hACE2-binding capability measured by IC50 of hACE2 against pseudoviruses. Error bars indicate mean ± s.d. of n = 5 biologically independent replicates. P-values were calculated using a two-tailed Student's $t$-test. *, $p < 0.05$; **, $p < 0.01$; ***, $p < 0.001$. No label on variants with $p > 0.05$. Variants with significantly stronger binding are coloured blue, while those with weaker binding are coloured red. All neutralization assays were conducted in at least two independent experiments.

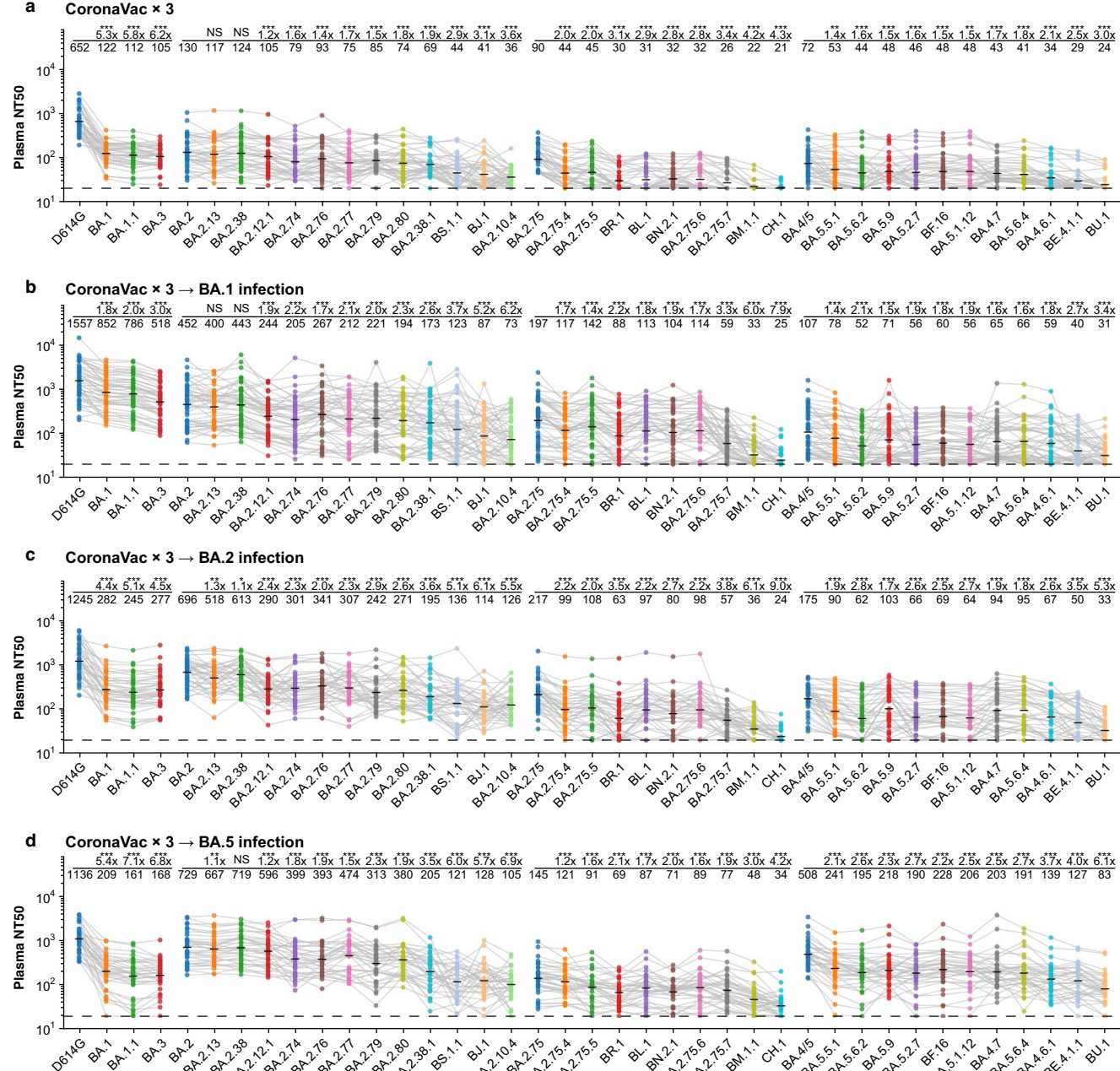

**Extended Data Fig. 3 | Plasma neutralization evasion of convergent Omicron variants. a-d,** NT50 against SARS-CoV-2 previous variants of concern and additional Omicron subvariants of plasma from vaccinated or convalescent individuals following breakthrough infection. Plasma samples, statistical methods and meaning of labels are the same as in Fig. 2. All neutralization assays were conducted in at least two independent experiments.

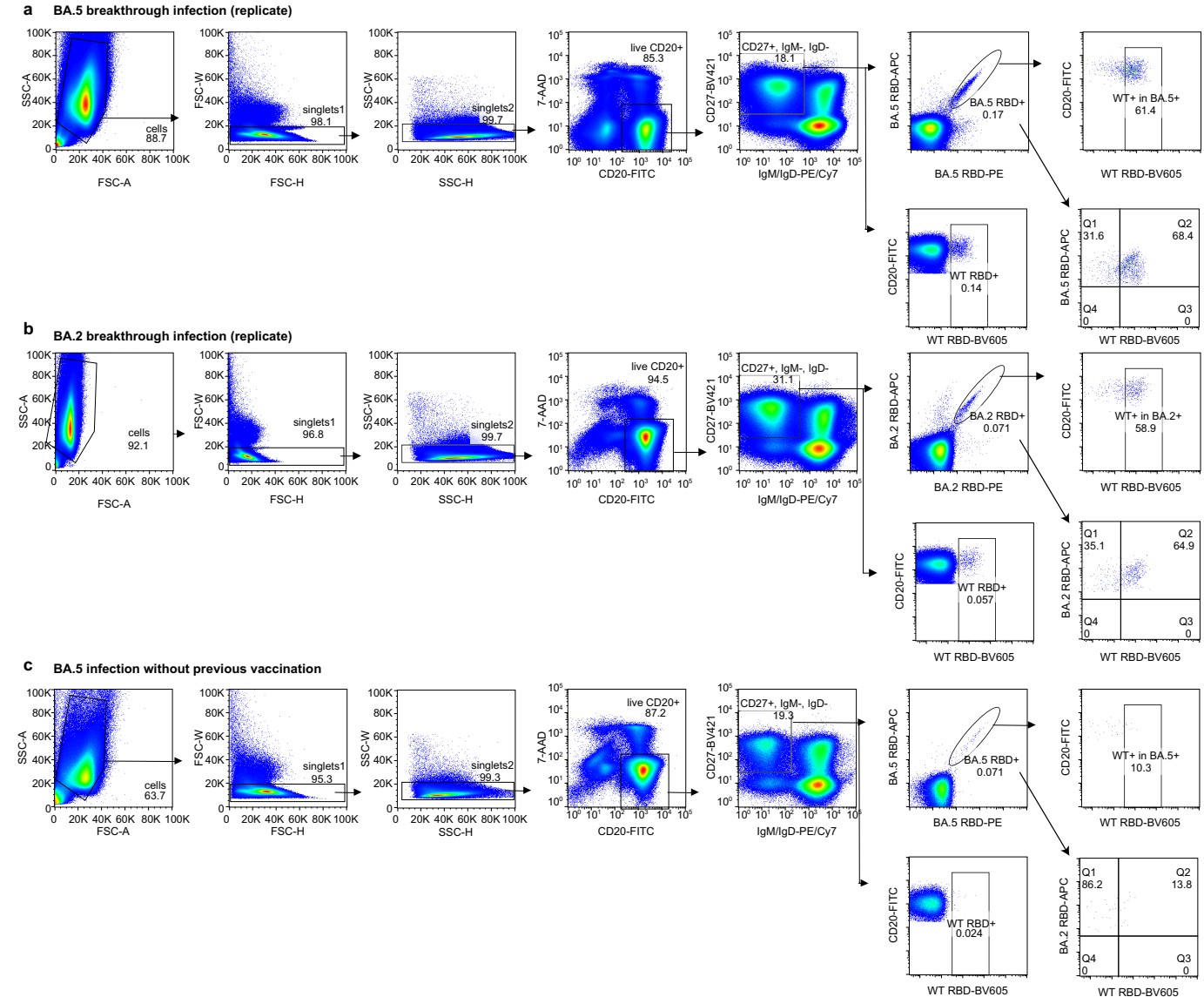

**Extended Data Fig. 4 | FACS gating strategy for isolating mAbs from BA.2 and BA.5 convalescent individuals. a**, FACS gating strategy of antigen-specific B cells from individuals who recovered from BA.5 breakthrough infection. Data from an independent experiment compared to Fig. 3a are shown here. **b**, FACS

gating strategy of antigen-specific B cells from individuals who recovered from BA.2 breakthrough infection. Data from an independent experiment compared to Fig. 3b are shown here. **c**, FACS gating strategy of antigen-specific B cells from individuals who recovered from BA.5 infection.

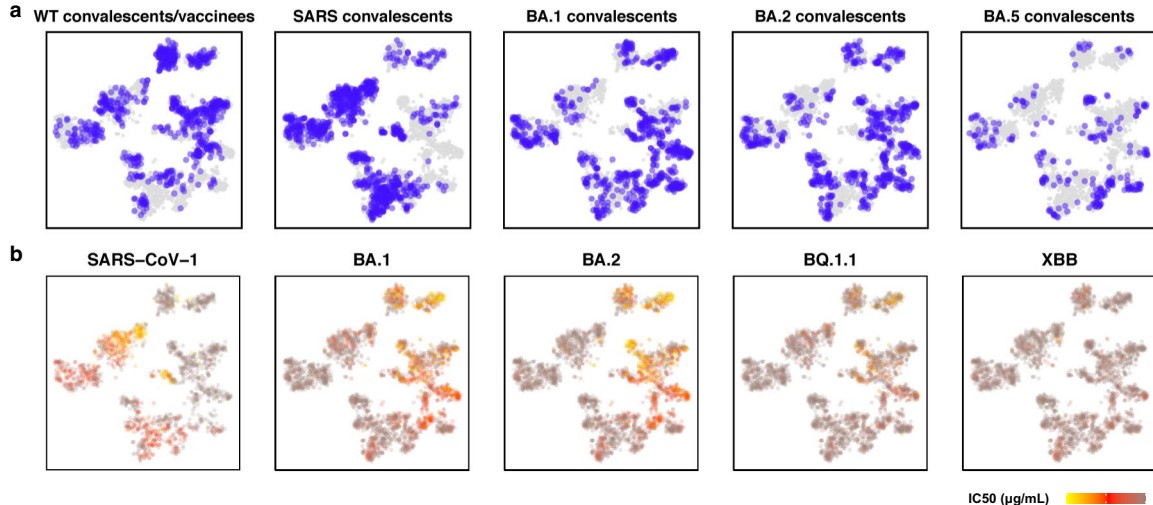

**Extended Data Fig. 5 | Distribution of antibody sources and neutralizing activities on the DMS landscape. a**, Sources of the 3051 mAbs involved in this study projected on the t-SNE of DMS profiles. **b**, IC50 against SARS-CoV-1 (N = 1870 determined), Omicron BA.1 (N = 3031), BA.2 (N = 3046), BQ.1.1 (N = 3051), and XBB (N = 3033) of these mAbs projected on the embedding. All neutralization assays were conducted in at least two independent experiments.

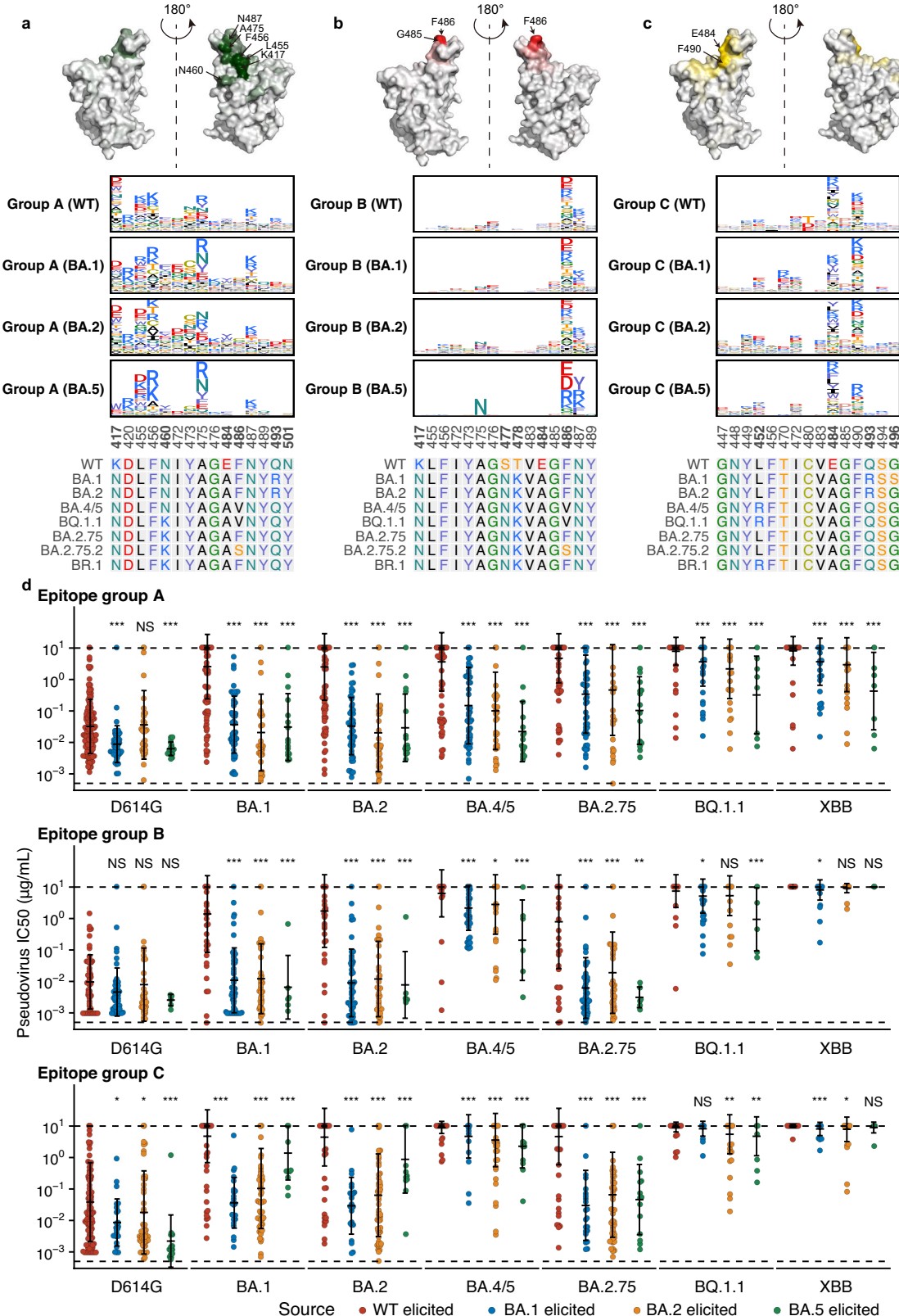

**Extended Data Fig. 6** | See next page for caption.

**Extended Data Fig. 6 | Escape hotspots and neutralization of mAbs in epitope groups A, B and C. a-c**, Average escape scores from DMS of epitope groups A (**a**), B (**b**), C (**c**) and each RBD residue. Scores are projected onto the structure of SARS-CoV-2 RBD (PDB: 6M0J). Average escape maps that indicate the score of each mutation from DMS on escape hotspots of antibodies, grouped by their sources, in epitope groups A (**a**), B (**b**) and C (**c**), and corresponding sequence alignment of SARS-CoV-2 WT and Omicron RBDs are also shown. The height of each amino acid in the escape maps represents its mutation escape score. Mutated sites in Omicron variants are marked in bold. **d**, Pseudovirus-neutralizing IC50 of antibodies in group A, B, and C, from wild-type vaccinated or convalescent individuals (WT-elicited, n = 133, 50, 106 for A-C, respectively), BA.1 (BA.1-elicited, n = 51, 49, 24), BA.2 (BA.2-elicited, n = 34, 36, 56) and BA.5 convalescent individuals (BA.5-elicited, n = 16, 6, 14). The geometric mean IC50s are labelled, and error bars indicate the geometric standard deviation. P-values are calculated using two-tailed Wilcoxon rank sum tests. *, $p < 0.05$; **, $p < 0.01$; ***, $p < 0.001$; NS, not significant, $p > 0.05$. All neutralization assays were conducted in at least two independent experiments.

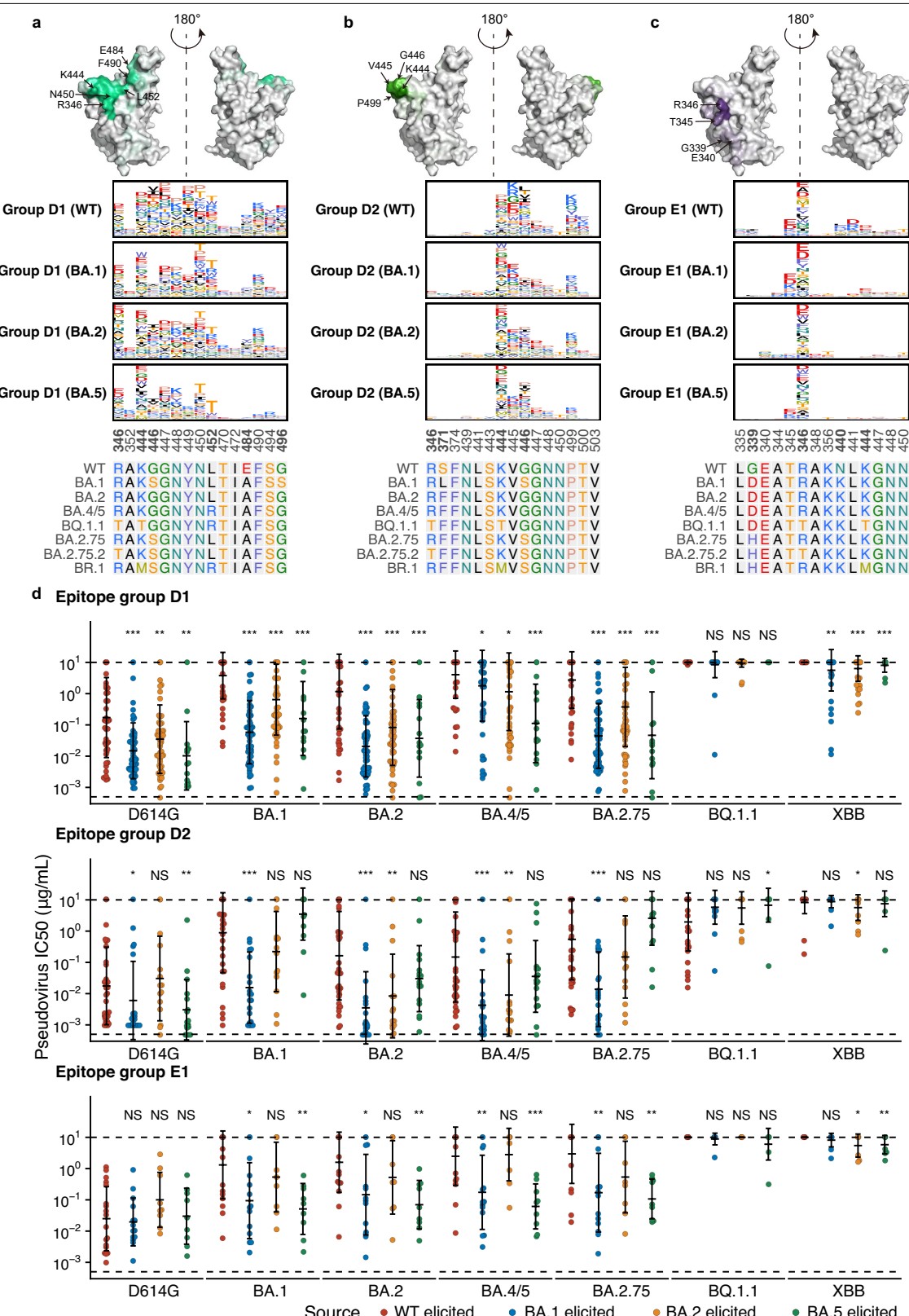

**Extended Data Fig. 7 | Escape hotspots and neutralization of mAbs in epitope group D and E1. a-c**, Average escape scores from DMS of epitope groups D1 (**a**), D2 (**b**), E1 (**c**) and each RBD residue. **d**, Pseudovirus-neutralizing IC50 of mAbs in group D1, D2, and E1 from WT vaccinated or convalescent individuals (n = 49, 37, 19 for D1, D2 and E1, respectively), BA.1 (n = 59, 21, 14 for D1, D2 and E1, respectively), BA.2 (n = 56, 15, 9 for D1, D2 and E1, respectively),

and BA.5 convalescent individuals (n = 14, 17, 9 for D1, D2 and E1, respectively). The geometric mean IC50s are labelled, and error bars indicate the geometric standard deviation. P-values are calculated using two-tailed Wilcoxon rank sum tests. *, p < 0.05; **, p < 0.01; ***, p < 0.001; NS, not significant, p > 0.05. All neutralization assays were conducted in at least two independent experiments.

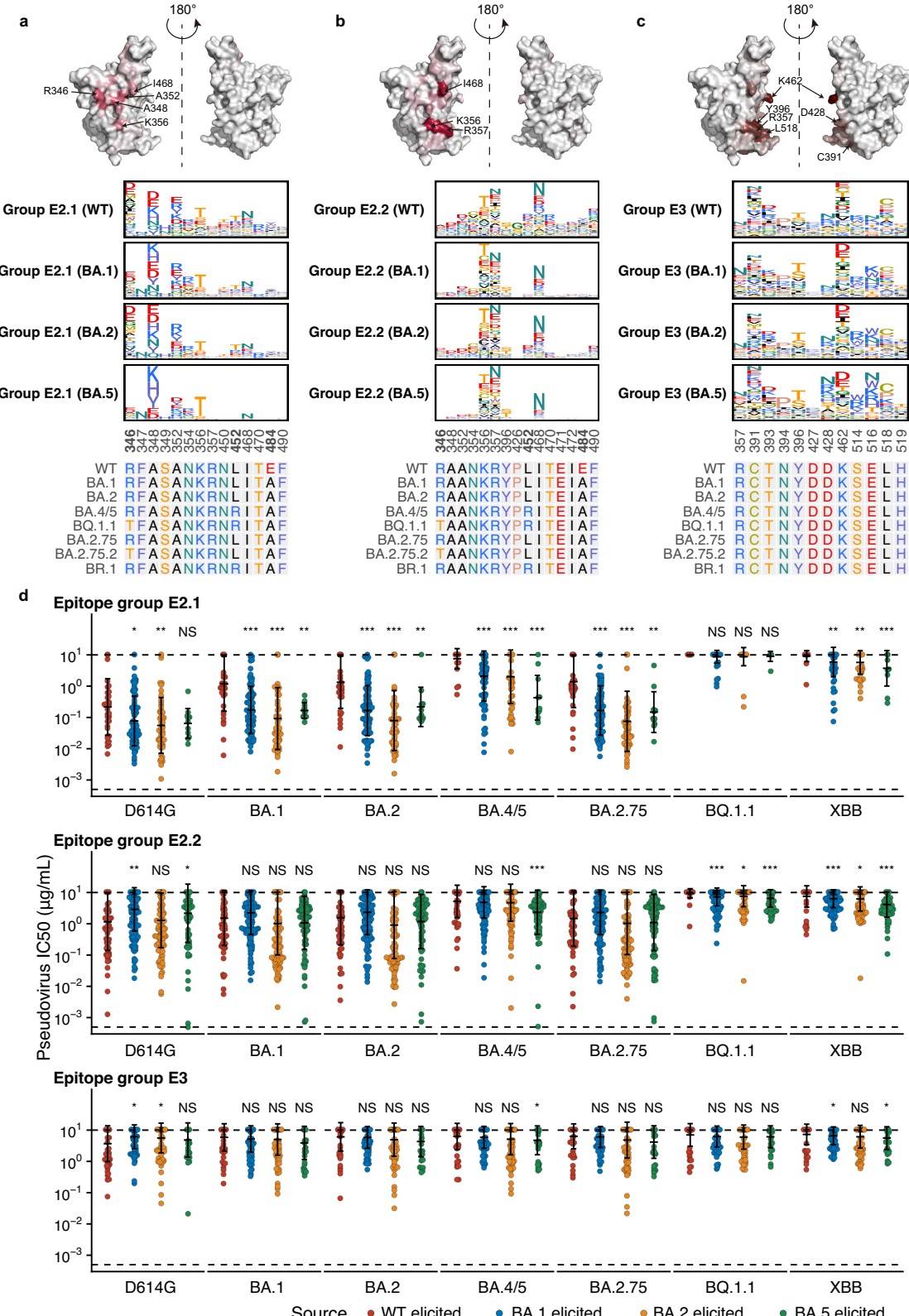

**Extended Data Fig. 8 | Escape hotspots and neutralization of mAbs in epitope group E2 and E3. a-c**, Average escape scores from DMS of epitope groups E2.1 (**a**), E2.2 (**b**), E3 (**c**) and each RBD residue. **d**, Pseudovirus-neutralizing IC50 of mAbs in group E2.1, E2.2, and E3 from WT vaccinated or convalescent individuals (n = 49, 37, 19 for E2.1, E2.2, and E3, respectively), BA.1 (n = 59, 21, 14 for E2.1, E2.2, and E3, respectively), BA.2 convalescents (n = 56, 15, 9 for E2.1, E2.2, and E3,

respectively), and BA.5 convalescent individuals (n = 14, 17, 9 for E2.1, E2.2, and E3, respectively). The geometric mean IC50s are labelled, and error bars indicate the geometric standard deviation. P-values are calculated using two-tailed Wilcoxon rank sum tests. *, p < 0.05; **, p < 0.01; ***, p < 0.001; NS, not significant, p > 0.05. All neutralization assays were conducted in at least two independent experiments.

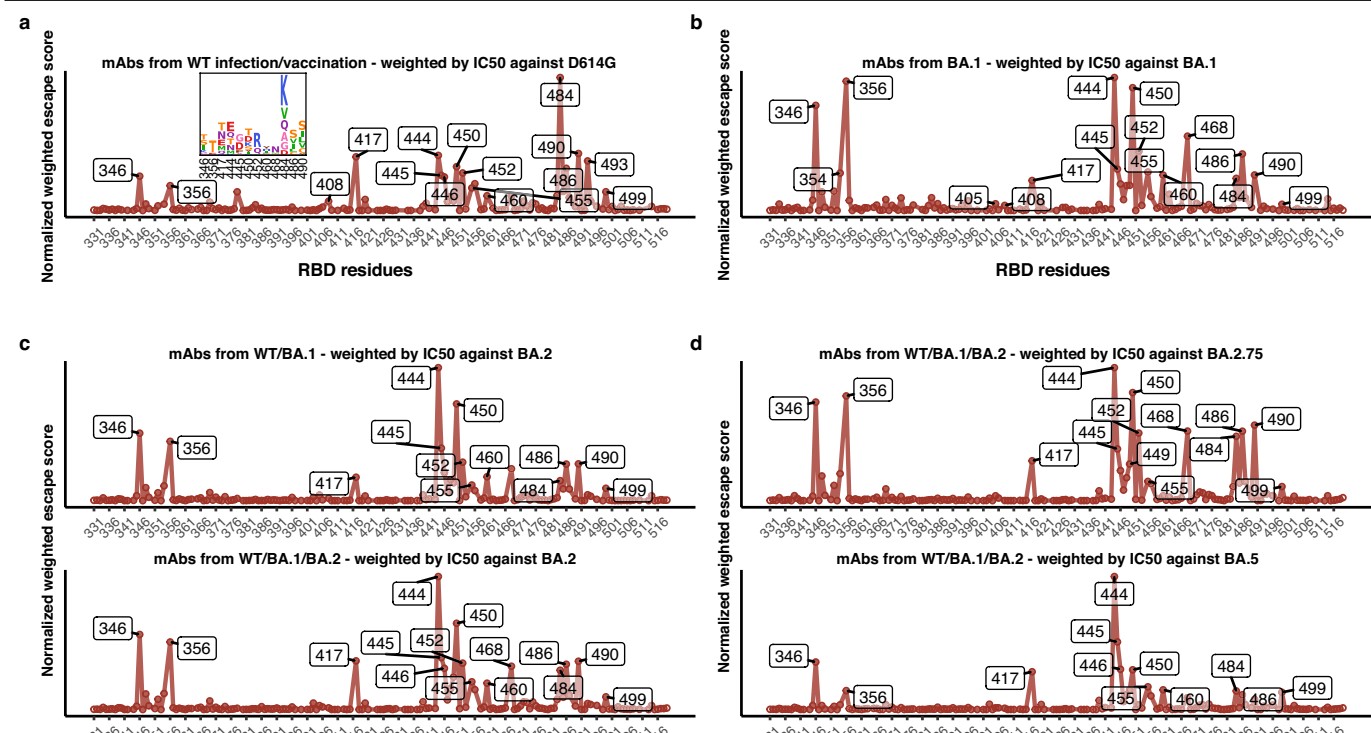

**Extended Data Fig. 9 | Predicted escape hotspots of SARS-CoV-2 variants.**
**a**, Normalized average escape scores weighted by IC50 against D614G using DMS profiles of mAbs from ancestral strain infection or vaccination with a logo plot showing specific mutations on important residues. **b**, Normalized average escape scores of mAbs from BA.1 breakthrough infection, weighted by IC50 against BA.1. **c**, Normalized average escape scores of mAbs from ancestral strain infection or vaccination and BA.1 breakthrough infection, weighted by IC50 against BA.2. **d**, WT/BA.1/BA.2-elicited mAbs with IC50 against BA.2.75 and BA.5, similar to Fig. 4b. All neutralization assays were conducted in at least two independent experiments.

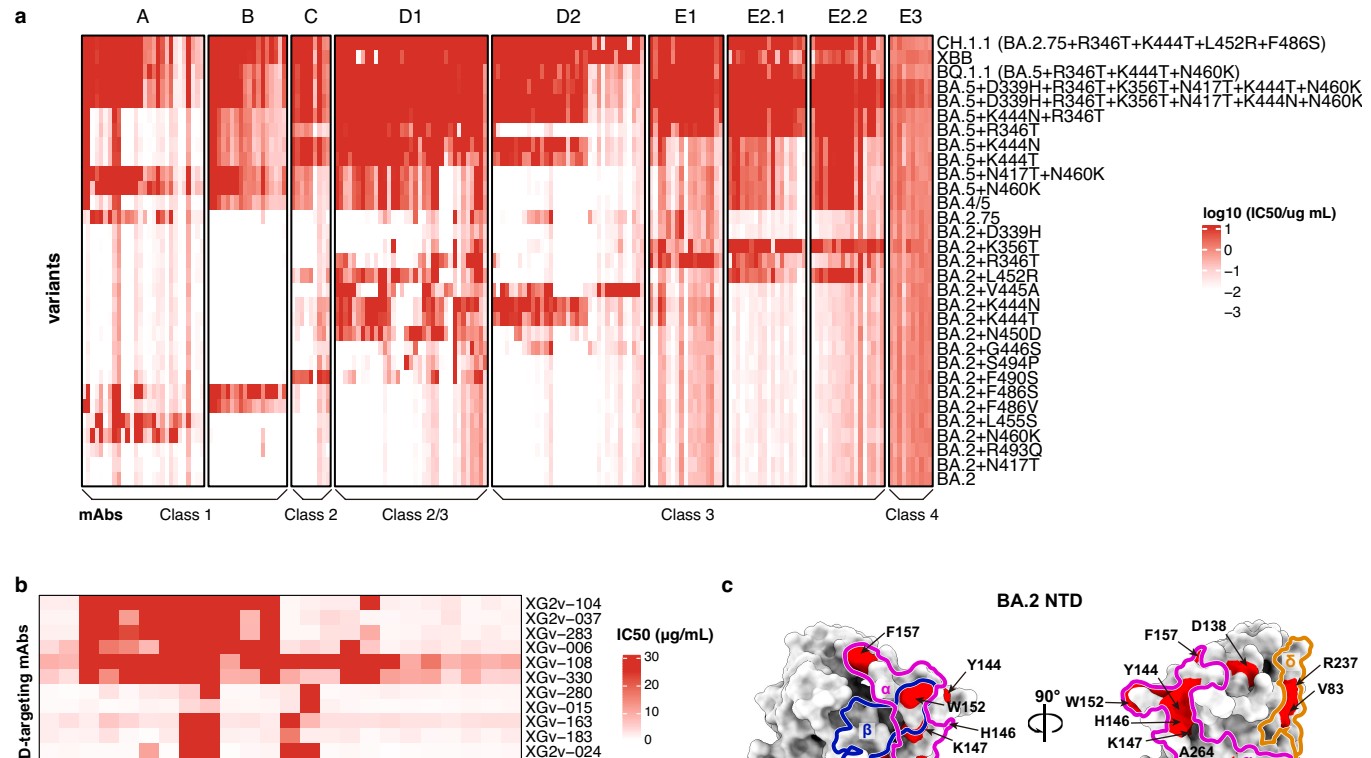

**a**

Columns (top): A | B | C | D1 | D2 | E1 | E2.1 | E2.2 | E3

variants (rows):
CH.1.1 (BA.2.75+R346T+K444T+L452R+F486S)
XBB
BQ.1.1 (BA.5+R346T+K444T+N460K)
BA.5+D339H+R346T+K356T+N417T+K444T+N460K
BA.5+D339H+R346T+K356T+N417T+K444N+N460K
BA.5+K444N+R346T
BA.5+R346T
BA.5+K444N
BA.5+K444T
BA.5+N417T+N460K
BA.5+N460K
BA.4/5
BA.2.75
BA.2+D339H
BA.2+K356T
BA.2+R346T
BA.2+L452R
BA.2+V445A
BA.2+K444N
BA.2+K444T
BA.2+N450D
BA.2+G446S
BA.2+S494P
BA.2+F490S
BA.2+F486S
BA.2+F486V
BA.2+L455S
BA.2+N460K
BA.2+R493Q
BA.2+N417T
BA.2

mAbs: Class 1 | Class 2 | Class 2/3 | Class 3 | Class 4

log10 (IC50/ug mL)
1
0
−1
−2
−3

**b**

NTD-targeting mAbs (rows):
XG2v−104
XG2v−037
XGv−283
XGv−006
XGv−108
XGv−330
XGv−280
XGv−015
XGv−163
XGv−183
XG2v−024
XGv−411
XG2v−025
XGv−378

columns:
BA.2
BA.4/5
BA.2.75
BU.1
BA.2.3.20
BA.2.10.4
BN.2.1
XBB
XBB.1
BA.2+K147E
BA.2+W152R
BA.2+Y144del
BA.2+V83A
BA.2+R237T
BA.2+F157L
BA.2+F146Q
BA.2+P251H
BA.2+D138N
BA.2+V213E
BA.2+A264T
BA.2+G257S
BA.2+Q183E
BA.2+I210T
BA.2+I210V

IC50 (µg/mL)
30
20
10
0

**c**

**BA.2 NTD**

(labels: F157, Y144, W152, H146, K147, A264, P251, Q183, I210, β, α, D138, R237, V83, G257, G213, δ, γ, 90°)

**Extended Data Fig. 10 | IC50 heatmaps of representative mAbs against constructed Omicron variants. a**, Colour shades indicate IC50 of antibodies (columns) against constructed Omicron BA.2 or BA.5 subvariants (rows) carrying mutations on the epitope of each group. The order of mAbs is the same as in Fig. 4c. **b**, IC50 of NTD-targeting antibodies against SARS-CoV-2 variants, which is related to Fig. 4e. **c**, Epitope groups and escape hotspots on BA.2 NTD. All neutralization assays were conducted in at least two independent experiments.

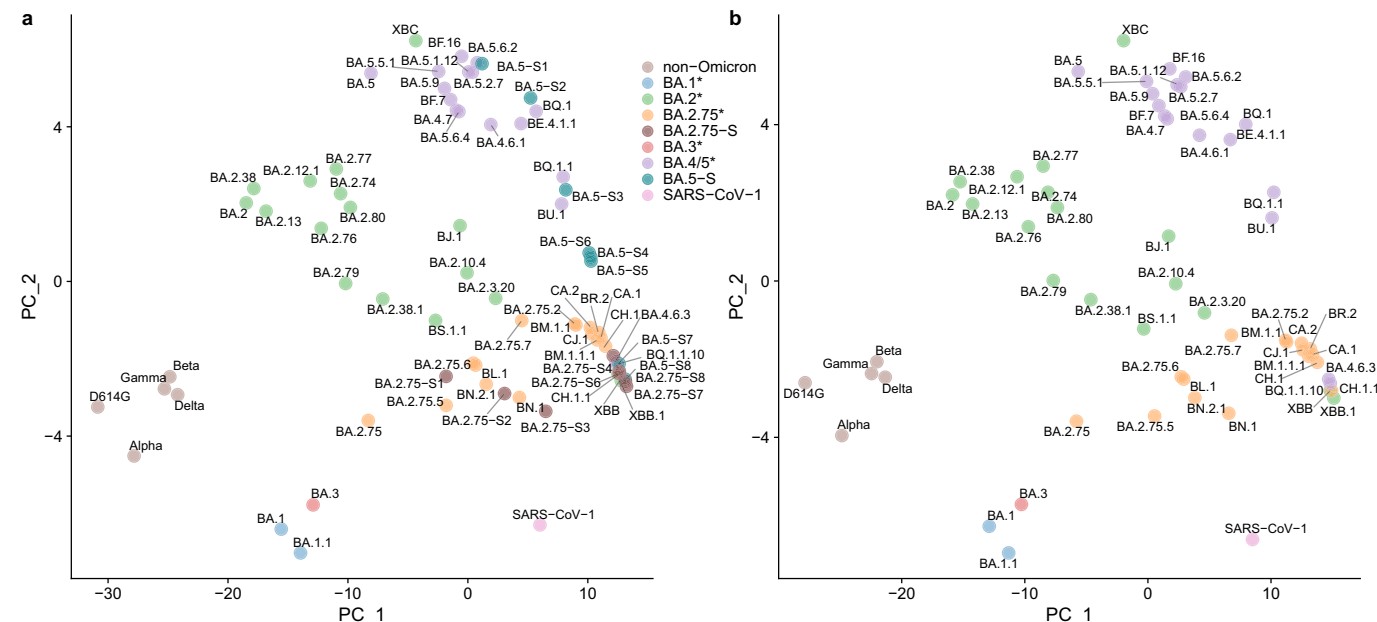

**Extended Data Fig. 11 | Antigenic map of current SARS-CoV-2 variants. a**, Antigenic map of SARS-CoV-2 variants constructed using plasma neutralization data by principal component analysis (PCA). **b**, Antigenic map of SARS-CoV-2 variants with constructed Omicron subvariants removed.

# Reporting Summary

## Statistics

For all statistical analyses, confirm that the following items are present in the figure legend, table legend, main text, or Methods section.

| n/a | Confirmed | |
|---|---|---|
| ☐ | ☒ | The exact sample size (*n*) for each experimental group/condition, given as a discrete number and unit of measurement |
| ☐ | ☒ | A statement on whether measurements were taken from distinct samples or whether the same sample was measured repeatedly |
| ☐ | ☒ | The statistical test(s) used AND whether they are one- or two-sided<br>*Only common tests should be described solely by name; describe more complex techniques in the Methods section.* |
| ☒ | ☐ | A description of all covariates tested |
| ☒ | ☐ | A description of any assumptions or corrections, such as tests of normality and adjustment for multiple comparisons |
| ☐ | ☒ | A full description of the statistical parameters including central tendency (e.g. means) or other basic estimates (e.g. regression coefficient) AND variation (e.g. standard deviation) or associated estimates of uncertainty (e.g. confidence intervals) |
| ☐ | ☒ | For null hypothesis testing, the test statistic (e.g. *F*, *t*, *r*) with confidence intervals, effect sizes, degrees of freedom and *P* value noted<br>*Give P values as exact values whenever suitable.* |
| ☒ | ☐ | For Bayesian analysis, information on the choice of priors and Markov chain Monte Carlo settings |
| ☒ | ☐ | For hierarchical and complex designs, identification of the appropriate level for tests and full reporting of outcomes |
| ☒ | ☐ | Estimates of effect sizes (e.g. Cohen's *d*, Pearson's *r*), indicating how they were calculated |

*Our web collection on statistics for biologists contains articles on many of the points above.*

## Software and code

Policy information about availability of computer code

| | |
|---|---|
| Data collection | Pseudovirus neutralization and ELISA data were collected by microplate spectrophotometer (PerkinElmer, HH3400).<br>FACS data was collected by Summit 6.0 (Beckman Coulter). |
| Data analysis | Neutralization assays data were analyzed using PRISM (v9.0.1) .<br>FACS data were analyzed by FlowJo 10.8.<br>Sequence alignment of Omicron sublineages was performed by biopython (v1.78); V(D)J sequence data were analyzed using Cell Ranger (v6.1.1), IgBlast (v1.17.1). Somatic hypermutation sites in the antibody variable domain were detected using Change-O toolkit (v1.2.0). Illumina barcodes sequencing data from deep mutational scanning experiments were analyzed using custom scripts (https://github.com/jianfcpku/SARS-CoV-2-RBD-DMS-broad) and Python package dms_variants (v0.8.9).<br>Custom scripts to analyze the escape mutation profiles data are available at https://github.com/jianfcpku/convergent_RBD_evolution.<br>Logo plots were generated by Python package logomaker (v0.8) and R package ggseqlogo (v0.1). For unsupervised clustering, we utilized Python package scipy (v1.7.0) and scikit-learn (v0.24.2) to perform multidimensional scaling (MDS), k-means clustering and t-Distributed<br>Stochastic Neighbor Embedding (t-SNE) embedding. 2D t-SNE plots are generated by ggplot2 (v3.3.3)<br>Multiple sequence alignments of sarbecovirus RBD were generated using ClustalOmega (v1.2.4) |

For manuscripts utilizing custom algorithms or software that are central to the research but not yet described in published literature, software must be made available to editors and reviewers. We strongly encourage code deposition in a community repository (e.g. GitHub). See the Nature Portfolio guidelines for submitting code & software for further information.

## Data

Policy information about availability of data

All manuscripts must include a data availability statement. This statement should provide the following information, where applicable:

- Accession codes, unique identifiers, or web links for publicly available datasets
- A description of any restrictions on data availability
- For clinical datasets or third party data, please ensure that the statement adheres to our policy

Processed mutation escape scores can be downloaded at https://github.com/jianfcpku/convergent_RBD_evolution. Sequences and neutralization of the antibodies are included in Supplementary Table 2. Raw sequencing data of DMS assays are available on China National GeneBank (db.cngb.org) with Project accession CNP0003808. We used vdj_GRCh38_alts_ensembl-5.0.0 as the reference of V(D)J alignment, which can be obtained from https://support.10xgenomics.com/single-cell-vdj/software/downloads/latest. We used PDB 6M0J for the structural model of SARS-CoV-2 RBD. The list of strains and the growth advantages were collected from the #24 collection of https://cov-spectrum.org. Designated lineages were from https://github.com/cov-lineages/pango-designation.

# Field-specific reporting

Please select the one below that is the best fit for your research. If you are not sure, read the appropriate sections before making your selection.

☒ Life sciences          ☐ Behavioural & social sciences          ☐ Ecological, evolutionary & environmental sciences

For a reference copy of the document with all sections, see nature.com/documents/nr-reporting-summary-flat.pdf

# Life sciences study design

All studies must disclose on these points even when the disclosure is negative.

| | |
|---|---|
| Sample size | A total of 3333 (3051 cross-reactive+282 Omicron-specific) antibodies were characterized in the manuscript. We analyzed all antibodies in hand and the sample size of antibodies in this study was sufficient to reach statistical significance by two-tailed binomial test for the differences in epitope distribution.<br>Plasma samples were obtained from 40 volunteers who received 3 doses of CoronaVac, 50 BA.1 breakthrough infection convalescent individuals, 39 BA.2 breakthrough infection convalescent individuals, and 36 BA.5 breakthrough infection convalescent individuals who all had received 3 doses of CoronaVac before infection. We analyzed all plasma samples collected and the sample size of plasma could reach statistical significance of NT50 values from neutralization assays by two-tailed Wilcoxon signed-rank test. We also used 4 primary BA.5 infection convalescent individuals' and 3 BA.2 primary infection convalescent individuals' PBMC samples in flow cytometry assay. No sample size calculation was performed. |
| Data exclusions | 846 antibodies were excluded from the study because of insufficient antibody or failed deep mutational scanning experiments, which is defined as no mutations scored two times of the median score. |
| Replication | Experimental assays were performed in at least two independent experiments according to or exceeding standards in the field. Specifically, we performed mutation screening using two independently constructed mutant libraries. We conducted all neutralization assays and ELISA in at least two independent experiments. All replicates for neutralization and ELISA are successful. |
| Randomization | Randomization was not required since we were applying a uniform set of measurements across the panel of monoclonal antibodies and plasma. As this is an observational study, randomization is not relevant. |
| Blinding | Blinding was not required since we were applying a uniform set of measurements across the panel of monoclonal antibodies and plasma. As this is an observational study, investigators were not blinded. |

# Reporting for specific materials, systems and methods

We require information from authors about some types of materials, experimental systems and methods used in many studies. Here, indicate whether each material, system or method listed is relevant to your study. If you are not sure if a list item applies to your research, read the appropriate section before selecting a response.

### Materials & experimental systems

| n/a | Involved in the study |
|---|---|
| ☐ | ☒ Antibodies |
| ☐ | ☒ Eukaryotic cell lines |
| ☒ | ☐ Palaeontology and archaeology |
| ☒ | ☐ Animals and other organisms |
| ☐ | ☒ Human research participants |
| ☒ | ☐ Clinical data |
| ☒ | ☐ Dual use research of concern |

### Methods

| n/a | Involved in the study |
|---|---|
| ☒ | ☐ ChIP-seq |
| ☐ | ☒ Flow cytometry |
| ☒ | ☐ MRI-based neuroimaging |

# Antibodies

| | |
|---|---|
| Antibodies used | ELISA: 0.25 μg/ml goat anti-human IgG(H+L)HRP (JACKSON, 109-035-003)<br>1 μg/ml H7N9 human IgG1 antibody HG1K (Sino Biologicals, Cat #HG1K) was used as negative control.<br>FACS: The cells were stained with FITC anti-human CD20 antibody (BioLegend, 302304), Brilliant Violet 421 anti-human CD27 antibody (BioLegend, 302824), PE/Cyanine7 anti-human IgM antibody (BioLegend, 314532), PE/Cyanine7 anti-human IgD antibody(BioLegend, 348210).<br>All human antibodies were expressed using Expi293F™ (Gibco, A14527)with codon-optimized cDNA and human IgG1 constant regions in house. The detailed sequence could be found in Supplementary material. |
| Validation | In this manuscript, we tested 3333 (3051 cross-reactive+282 Omicron-specific) human IgG1 antibodies. All antibodies were expressed using Expi293F™ with codon-optimized cDNA and human IgG1 constant regions. All antibodies' species and specificity to RBD were validated by ELISA. All antibodies neutralization ability was verified by VSV-based pseudotyped virus assays. Details and sequences for all SARS-CoV-2 antibodies evaluated in this study is included in Supplementary Table.<br>Goat anti-human IgG(H+L)HRP (JACKSON, 109-035-003): Based on immunoelectrophoresis and/or ELISA, the antibody reacts with whole molecule human IgG. It also reacts with the light chains of other human immunoglobulins. No antibody was detected against non-immunoglobulin serum proteins. The antibody may cross-react with immunoglobulins from other species.<br>FITC anti-human CD20 antibody was validated by successful staining and FC analysis according to the manufacturer's website https://www.biolegend.com/en-us/products/fitc-anti-human-cd20-antibody-558 and previous publication: Mishra A, et al. 2021. Cell 184(13):3394-3409.e20<br>Brilliant Violet 421 anti-human CD27 antibody was validated by successful staining and FC analysis according to the manufacturer's website https://www.biolegend.com/en-us/products/brilliant-violet-421-anti-human-cd27-antibody-7276 and previous publication Dugan HL, et al. 2021. Immunity. 54(6):1290-1303<br>PE/Cyanine7 anti-human IgM antibody was validated by successful staining and FC analysis according to the manufacturer's website https://www.biolegend.com/en-us/products/pe-cyanine7-anti-human-igm-antibody-12467 and previous publication: Shehata L, et al 2019. Nat Commun. 10:1126<br>PE/Cyanine7 anti-human IgD antibody was validated by successful staining and FC analysis according to the manufacturer's website https://www.biolegend.com/en-us/products/pe-cyanine7-anti-human-igd-antibody-6996 and previous publication: Ahmed R et al. 2019. Cell. 177(6):1583-1599. |

# Eukaryotic cell lines

Policy information about cell lines

| | |
|---|---|
| Cell line source(s) | Monoclonal antibody expression: Expi293F™ (Gibco, A14527);<br>Yeast display: EBY100 (ATCC MYA-4941);<br>Pseudutyped virus neutralization assay: Huh-7 (JCRB 0403) ;<br>293T(ATCC, CRL-3216); |
| Authentication | No authentication was performed beyond manufacturer standards; |
| Mycoplasma contamination | Not tested for mycoplasma contamination; |
| Commonly misidentified lines<br>(See ICLAC register) | No commonly misidentified cell lines were used in the study. |

# Human research participants

Policy information about studies involving human research participants

| | |
|---|---|
| Population characteristics | Samples were obtained from 40 volunteers who received 3 doses of CoronaVac, 50 BA.1 breakthrough infection convalescent individuals, 39 BA.2 breakthrough infection convalescent individuals, and 36 BA.5 breakthrough infection convalescent individuals who all had received 3 doses of CoronaVac before infection. PBMC samples from 4 primary BA.5 infection convalescent individuals and 3 BA.2 primary infection convalescent individuals were also used in in flow cytometry assay. Gender, age, vaccination profiles, and sampling time point are described in Supplementary table 1. Breakthrough infection individuals are listed in sheet1, and primary infection (without vaccination) samples are listed in sheet2.<br>For the BA.1 breakthrough infection cohort, we presume all individuals were infected by BA.1 since these individuals were infected during the BA.1 wave in Tianjin, China in Jan 2022. A total of 430 patients were confirmed BA.1-infected and no other lineages were detected by sequencing.<br>For the BA.2 breakthrough infection cohort, we presume all individuals were infected by BA.2 since these individuals were infected during the BA.2 wave in Beijing, China in March-May 2022. Some of them were confirmed with sequencing. Others are epidemiologically linked to the confirmed patients.<br>For the BA.5 breakthrough infection cohort, we presume all individuals were infected by BA.5 since some of these individuals were confirmed with sequencing and others are epidemiologically linked to the confirmed patients. |
| Recruitment | Patients were recruited on the basis of CoronaVac vaccination, post-vaccination BA.1, BA.2 or BA.5 infection, and primary infection of BA.2/BA.5 without vaccination. The only extrusion criteria used were HIV or other debilitating disease. The time intervals between sampling and hospital admission of BA.5 infection samples are shorter than that of BA.1 and BA.2 convalescents, which may cause potential influence on humoral immune responses. |

| Ethics oversight | Samples from vaccinees and individuals who had recovered from BA.1, BA.2, or BA.5 infection were obtained under study protocols approved by Beijing Ditan Hospital, Capital Medical University (Ethics committee archiving No. LL-2021-024-02) and the Tianjin Municipal Health Commission, and the Ethics Committee of Tianjin First Central Hospital (Ethics committee archiving No. 2022N045KY). All donors provided written informed consent 485 for the collection of information, the use of blood and blood components, and publication of data generated from this study. |
|---|---|

Note that full information on the approval of the study protocol must also be provided in the manuscript.

# Flow Cytometry

## Plots

Confirm that:

☒ The axis labels state the marker and fluorochrome used (e.g. CD4-FITC).

☒ The axis scales are clearly visible. Include numbers along axes only for bottom left plot of group (a 'group' is an analysis of identical markers).

☒ All plots are contour plots with outliers or pseudocolor plots.

☒ A numerical value for number of cells or percentage (with statistics) is provided.

## Methodology

| Sample preparation | Whole blood sample were diluted 1:1 with PBS+2% FBS (Gibco) and subjected to Ficoll (Cytiva) gradient centrifugation. Plasma was collected from upper layer. Cells were collected at the interface and further prepared by centrifugation, red blood cells lysis (Invitrogen eBioscience) and washing steps. Samples were stored in FBS (Gibco) with 10% DMSO (Sigma) in liquid nitrogen if not used for downstream process immediately. Cryopreserved PBMCs were thawed in PBS+2% FBS. CD19+ B cells were isolated from PBMCs with EasySep Human CD19 Positive Selection Kit II (STEMCELL, 17854). Every 10^6 B cells in 100 µl solution were stained with 3 µl FITC antihuman CD20 antibody (BioLegend, 302304, clone: 2H7), 3.5 µl Brilliant Violet 421 anti-human CD27 antibody (BioLegend, 302824, clone: O323), 2 µl PE/Cyanine7 anti-human IgM antibody (BioLegend, 314532, clone: MHM-88), 2 µl PE/Cyanine7 anti-human IgD antibody (BioLegend, 348210, clone: IA6-2), 0.13 µg biotinylated SARS-CoV-2 BA.2 RBD protein (customized from Sino Biological) or 0.13 µg biotinylated SARS-CoV-2 BA.5 RBD protein (customized from Sino Biological) conjugated with PE-streptavidin (BioLegend, 405204) or APC-streptavidin (BioLegend, 405207), 0.13 µg SARS-CoV-2 WT biotinylated RBD protein 500 (Sino Biological, 40592-V27H-B) conjugated with Brilliant Violet 605 Streptavidin (BioLegend, 405229). Cells are also labeled with biotinylated RBD conjugated to DNA-oligostreptavidin. Cells were washed twice after 30 minutes incubation on ice. 7-AAD (Invitrogen, 00-6993-50) were used to label dead cells. |
|---|---|
| Instrument | Moflo Astrios EQ (BeckMan Coulter) |
| Software | Summit 6.0 (Beckman Coulter) for cell sorting; FlowJo 10.8 for data analysis. |
| Cell population abundance | BA.2 infecion without history infection: 7AAD-&CD20+/singletes=65.7%, CD27+&IgM-&IgD-/7AAD-&CD20+=19.1%, BA.2-RBD+/CD27+&IgM-&IgD-=0.048%, WT-RBD+/BA.2-RBD+= 19.7%<br><br>BA.2 breakthrough infection : 7AAD-&CD20+/singletes=85.9%, CD27+&IgM-&IgD-/7AAD-&CD20+=20.2%, BA.2-RBD+/CD27+&IgM-&IgD-=0.086%, WT-RBD+/BA.2-RBD+= 71.1%<br>BA.2 breakthrough infection (replicate) : 7AAD-&CD20+/singletes=94.5%, CD27+&IgM-&IgD-/7AAD-&CD20+=31.1%, BA.2-RBD+/CD27+&IgM-&IgD-=0.071%, WT-RBD+/BA.2-RBD+= 58.9%<br><br>BA.5 breathrough infecion: 7AAD-&CD20+/singletes=85.8%, CD27+&IgM-&IgD-/7AAD-&CD20+=16.9%, BA.5-RBD+/CD27+&IgM-&IgD-=0.28%, WT-RBD+/BA.5-RBD+= 68.6%<br>BA.5 breakthrough infection (replicate): 7AAD-&CD20+/singletes=85.3%, CD27+&IgM-&IgD-/7AAD-&CD20+=18.1%, BA.5-RBD+/CD27+&IgM-&IgD-=0.17%, WT-RBD+/BA.5-RBD+= 61.4%<br><br>BA.5 infecion without history infection: 7AAD-&CD20+/singletes=87.2%, CD27+&IgM-&IgD-/7AAD-&CD20+=19.3%, BA.5-RBD+/CD27+&IgM-&IgD-=0.071%, WT-RBD+/BA.5-RBD+= 10.3% |
| Gating strategy | 7-AAD−CD20+CD27+IgM−IgD− SARS-CoV-2 BA.2 RBD+ or BA.5+ cells were sorted. The detailed FSC/SSC gating scheme is showed in Extended Data Figure 4. |

☒ Tick this box to confirm that a figure exemplifying the gating strategy is provided in the Supplementary Information.

