## [Peer Review File · Nature]

Manuscript Title: Imprinted SARS-CoV-2 humoral immunity induces convergent Omicron RBD evolution

Reviewer Comments & Author Rebuttals

Reviewer Reports on the Initial Version:

Referees' comments:

Referee #1 (Remarks to the Author):

This is a superb and important paper. It simultaneously accomplishes three things:

1. Being the state-of-the-art characterization of neutralization escape of current SARS-CoV-2 variants.
2. Providing a rich map of future escape mutations that can be used to understand ongoing evolution.
3. Insight into the immunological response to repeated boosting.

As a pre-print this paper has already substantially shaped thinking about new variants and their evolution, and in a sense has already undergone extensive pre-publication review on Twitter and elsewhere. I recommend the paper for acceptance, with the following suggested minor revisions:

- Over the last few weeks, the authors have added in their Twitter posts and to the bioRxiv pre-print details on some new variants like XBB. It would be great if that could be included in this version.
- It should be more clearly emphasized that the B cells and serum are generally collected ~1 month post breakthrough. There is some evidence (eg, Kaku et al, <https://www.biorxiv.org/content/10.1101/2022.09.21.508922v1.abstract>) that there can be some affinity maturation that leads to better neutralization by Wuhan-Hu-1 specific antibodies at later time points. This fact should be mentioned.
- This study uses the CoronaVac vaccine and finds ~80% of B-cells after breakthrough infection are Wuhan-Hu-1 reactive and so presumably boosted from earlier exposures. How does this compare to other vaccines? My impression is that it is even higher for mRNA vaccines?
- While I agree with the conclusions, it is important to note that BA.5 breakthrough does provide higher serum titer to BA.5 and also BA.5 derived variants like BQ.1.1 than does breakthrough with BA.2 and especially the more distant BA.1. This fact does suggest that there is some useful variant specific boosting.
- The work on forecasting new mutations and generating the synthetic escape mutants is interesting. This work is all done in the context of VSV pseudotyping, and studies a virus (SARS-CoV-2) that is already circulating and evolving rapidly in humans. Therefore, I would consider this work to be safe itself (due to use of VSV), and not overly risk: it is not using a potential pandemic virus (like a bat

CoV), and is exploring mutations that are likely to arise naturally over the next 6-12 months of virus evolution, and whose study has direct applications for vaccine and antibody development. Nonetheless, given the controversy around other experiments that for instance have involved direct engineering of actual potential pandemic viruses (not pseudoviruses, not already circulating in humans), I think a few sentences should be added explaining the justification for this work and the reasons why the benefits would seem to outweigh the risks.

- I agree with the overall points, but do think that some aspects of the interpretation are a bit too dire. The history of other rapidly evolving viruses (influenza, coronaviruses) suggests that people still make neutralizing antibodies to new strains late into their lives, probably due to the combined action of new B cells and affinity maturation. So perhaps the way things are framed could also take that into account.

- How many donors did the B cells come from, and how consistent were results among donors?

Referee #2 (Remarks to the Author):

This study sets out to investigate the impact and driving force behind convergent mutations in Omicron BA.2 and BA.5 variants. In terms of impact they use pseudotype viruses containing Spike of BA.2 and BA.5 variants and examined how they were neutralized by (1) therapeutic antibodies, or (2) by antibodies in sera collected post dose 3 of vaccine or post breakthrough infection with BA.1, BA.2 or BA.5. This shows that variants are relatively poorly neutralized by most therapeutic mAbs and post-exposure sera. In terms of driving forces they characterize the epitope repertoire of B cell derived recombinant mAbs from vaccinees who had breakthrough infections with BA.2 and BA.5 using deep mutational scanning. This shows a skewing towards non-neutralizing epitopes, particularly after BA.5 infection compared to WT exposure. Additionally, the epitope repertoire of neutralizing Abs (nAbs) derived from post-breakthrough infection RBD-reactive B cells is relatively narrow compared to that of equivalent B cells detected post vaccination or infection with ancestral/WT/Wuhan strain. Although this is relatively clearly demonstrated, the assertion that the driving force for convergent mutations is immune imprinting comes across as circumstantial rather than directly demonstrated. While most RBD reactive B cells detected post breakthrough infection are Wuhan strain cross-reactive, there is limited evidence to show that a different pattern is observed in the absence of prior exposure to Wuhan strain. Finally, the authors investigate how BA.2 and BA.5 viruses engineered to contain combinations of the convergent substitutions are neutralized by sera from participants with breakthrough infections and find that four substitutions at positions 346, 444, 452, and 486 are sufficient to confer substantial escape.

The study presents important original results that are of importance for the design of vaccines and therapeutics

The data and methodology are of high quality and validity with the exception that there is no description of how the ACE2 binding affinity of RBD variants was measured in Fig 2b, 4 etc. In past papers the authors have used surface plasmon resonance (SPR) but here they use ELISA. This should be described and ideally validated by comparison to SPR.

The statistics used appear to be appropriate with the exception that in Extended Data Figs 5, 6 & 7d: it is not clear why the statistical comparisons are pairwise for adjacent exposure groups rather than

comparing all to the group only exposed to WT virus?

Major Comments

Abstract: is long, and lacks clarity so that it is difficult to discern the main aims and findings that: (1) nAb responses converge on a relatively restricted range of epitopes after breakthrough infection with variant strains as compared to initial exposure to WT strains, and (2) that antibodies become increasingly directed towards non-neutralizing epitopes after these breakthrough infections.

The statement “Additionally, the precise convergent RBD mutations and evolution trends of BA.2.75/BA.5 subvariants could be inferred by integrating the neutralization-weighted DMS profiles of mAbs from various immune histories (3051 mAbs in total).” is not a study finding.

The statement “Moreover, we demonstrated that as few as five additional convergent mutations based on BA.5 or BA.2.75 could completely evade most plasma samples, including those from BA.5 breakthrough infection, while retaining sufficient hACE2-binding affinity.” differs from line 309 which suggests four substitutions are sufficient.

Line 141: “Similar to that reported in BA.1 breakthrough infection, immune imprinting, or so-called “original antigenic sin”, is also observed in BA.2 and BA.5 breakthrough infection 2,36-39. Post-vaccination infection with BA.2 and BA.5 mainly recalls cross-reactive memory B cells elicited by wildtype-based vaccine, but rarely produces BA.2/BA.5 specific B cells, similar to BA.1 breakthrough

Referee #3 (Remarks to the Author):

A. Cao et al. combine sequence analysis, neutralization sensitivity, ACE-2 sensitivity, and modeling of deep mutational scanning to discern immune pressure being placed on SARS-CoV-2 and how it escapes from neutralization. Overall, it is an excellent study that makes significant points for vaccine design and for understanding SARS-CoV-2 humoral immunity. The sequence analysis in the manuscript shows the presence of shared mutations in different omicron lineages. The authors also show that monoclonal antibodies and vaccinee sera struggle to neutralize the newly emerging variants, but it should be stated in the text that most of the titers only change by 3-6 fold. The authors then isolate ~1700 monoclonal antibodies and show how the antibody repertoire shifts in epitope specificity after omicron infection and that many of the responding B cells are cross-reactive with earlier Wuhan-Hu-1 sequence RBD. The authors then use deep mutational analyses coupled with neutralizing activity of the mAbs to identify sites that could mutate to escape the neutralizing antibodies. These potential mutational sites could be useful for defining the next variants to emerge. Generating pseudoviruses with the potential mutation combinations resulted viruses that fully or nearly completely escaped vaccinee sera. In total, this work would indicate SARS-CoV-2 can completely escape from vaccine antibody neutralization responses without compromising ACE-2 binding. The concept of original antigenic sin or antibody imprinting is of concern and the authors show at the monoclonal antibody level how the humoral response to BA.1, BA.2, and BA.5 infection shifts relative to what happens with Wuhan-Hu-1 infection. This shift in epitopes recognized by omicron-induced antibodies is problematic based on the authors demonstration that the sublineages are continuing to accumulate escape mutations and broader epitope specificity will be needed to maintain neutralization activity against the newly emerging subvariants. Cao and colleagues define plausible mutations that could accumulate in SARS-CoV-2 and show these viruses are completely resistant to neutralization by vaccinee sera. Overall, the references and statistical analyses seem appropriate. The results support the conclusion. Instances where the authors

speculate on future events are noted below and suggested to be moved to the discussion.

Major comments:

Figure 1: The first point made by the authors is that Omicron sublineages contain the same mutations suggesting convergent mutational patterns in Spike relative to the whole genome. Meaning that while viral genomes place subvariants into different clades in a phylogenetic tree, the spike sequences include several common amino acid changes. How significant is this result? This reviewer believes this observation has been seen with earlier variants from completely different PANGO lineages (for example E484K, L452R).

1. A phylogenetic tree of spike sequences from each of the viruses shown in 1a would help the reader understand the extent of convergent spike sequences that come from different sublineages.
2. If one posits that the sublineages are evolving in the same geographic region, then immune pressure may account for similar spike mutations. Have you examined the presence of these mutations by geographic location?
3. How many sublineages in BA.4/5 must have a mutation observed in BA.2 for the mutation to be considered shared with BA.2? Is finding the mutation in one BA.4/5 sequence considered convergent evolution. Providing a threshold would help the reader.
4. Is convergent evolution only present in BA.2, BA.2.75, BA.5? If you compared Delta and Kappa sublineages would you see the same phenomenon?

Figure 2:

While binding affinity and neutralization inhibitory concentration are correlated they are not the same parameter. Please replace ACE2 binding affinity with ACE2 inhibition or ACE2 inhibitory concentration.

1. Line 76 mentions BA.5.5.1 being resistant, but I think BA.5.1.12 was meant to be mentioned since it is more resistant than BA.5.5.1.
2. Line 110: Please delete "they may tolerate even more escape mutations" as this statement is not shown with data and is only speculation.
3. Lines 126-130 are interpretation of the results and would be more appropriate in the discussion.

Figure 3:

Could you speculate in the discussion on whether germinal center B cells in lymph nodes would have the same cross-reactivity and epitope shift. Could germinal center B cells that are responding to omicron infection or vaccination show different specificity and mutation percentages. It is plausible that at the time you looked the new antibody responses was still ongoing.

Figure 3d: From how many BA.2 and BA.5 convalescent individuals were the antibodies isolated.

Minor comments:

Figure 2a, 5b: Define what the symbol in the table means. I assume it means no neutralization.

Figure 2a: The color scheme will be difficult for red-green color blind readers. It may also be better to color squares with titers between 100-1000 since white usually means no neutralization.

Figure 4c: labels indicating virus is shown in rows and antibodies are shown in columns would be

helpful.

Line 346: I do not think the use of revoke conveys the intended meaning of the sentence. Perhaps, invoke, recall, engage, or re-stimulate would be better word choices.

Author Rebuttals to Initial Comments:

Response to the referees

Referee #1 (Remarks to the Author):

This is a superb and important paper. It simultaneously accomplishes three things:

1. Being the state-of-the-art characterization of neutralization escape of current SARS-CoV-2 variants.
2. Providing a rich map of future escape mutations that can be used to understand ongoing evolution.
3. Insight into the immunological response to repeated boosting.

Response: Thanks for your recognition of our work.

As a pre-print this paper has already substantially shaped thinking about new variants and their evolution, and in a sense has already undergone extensive pre-publication review on Twitter and elsewhere. I recommend the paper for acceptance, with the following suggested minor revisions:

- Over the last few weeks, the authors have added in their Twitter posts and to the bioRxiv pre-print details on some new variants like XBB. It would be great if that could be included in this version.

Response: We have included the data about the new variants, including XBB, XBB.1, CA.1, CA.2, CH.1.1, BA.4.6.3, BQ.1.18 and some other concerning variants (Fig. 2 and Extended Data Fig. 1-2 in the revised manuscript).

- It should be more clearly emphasized that the B cells and serum are generally collected ~1 month post breakthrough. There is some evidence (eg, Kaku et al, <https://www.biorxiv.org/content/10.1101/2022.09.21.508922v1.abstract>) that there can be some affinity maturation that leads to better neutralization by Wuhan-Hu-1 specific antibodies at later time points. This fact should be mentioned.

Response: Thanks for the important suggestion. We emphasized this point in the revised text (Line 275). We also discussed the potential effects of the waning immunity and affinity maturation on neutralization and memory B cells (Line 347-352).

- This study uses the CoronaVac vaccine and finds ~80% of B-cells after breakthrough infection are Wuhan-Hu-1 reactive and so presumably boosted from earlier exposures. How does this compare to other vaccines? My impression is that it is even higher for mRNA vaccines?

Response: This is an important point. However, we could not recruit enough participants who received mRNA vaccines and experienced Omicron breakthrough infections, which is a limitation

of the study. Given that most mRNA vaccines, including BNT162b2 and mRNA-1273, used mRNAs encoding WT Spike glycoprotein as the immunogen, we believe there would be a similar phenomenon in mRNA-vaccinated individuals. A recent report of pseudovirus neutralization results using serum from BA.4/5 convalescents with previous mRNA vaccination suggested similar immune imprinting patterns (<https://www.biorxiv.org/content/10.1101/2022.10.22.513349v1>). Another two studies that analyzed the memory B cells from BA.1 infection or boosting with previous mRNA vaccination also got similar results, and indeed mRNA-vaccinated individuals displayed an even higher proportion of cross-reactive memory B cells, which may be contributed by the overall stronger humoral immune response of mRNA vaccines compared to inactivated vaccines (<https://doi.org/10.1101/2022.09.21.508922>, <https://doi.org/10.1101/2022.09.22.509040>). We also discussed these related works in the discussion (Line 340-346).

- While I agree with the conclusions, it is important to note that BA.5 breakthrough does provide higher serum titer to BA.5 and also BA.5 derived variants like BQ.1.1 than does breakthrough with BA.2 and especially the more distant BA.1. This fact does suggest that there is some useful variant specific boosting.

Response: We agree that BA.5 boosters will induce better neutralization against BA.5-derived convergent variants. Noteworthy, the increased neutralization may be greatly contributed by NTD-targeting NAbs, which are enriched in BA.2/BA.5 convalescents (<https://www.science.org/doi/10.1126/sciimmunol.ade2283>). These NTD-targeting NAbs could be easily escaped by many BA.2.75-derived variants and BA.5-derived variants with specific mutations on NTD (See Fig. 5g, comparing BA.5-S6 with BA.5-S7). Such mutations, especially Y144del, have rapidly emerged in the latest BA.5-derived variants, such as BA.4.6.3 (included in Fig. 2), BQ.1.8 and BQ.1.18. Therefore, we should pay close attention to these emerging variants and continuously evaluate the efficacy of BA.5-based boosters against them. We have discussed this in the revised manuscript (Line 353-361).

- The work on forecasting new mutations and generating the synthetic escape mutants is interesting. This work is all done in the context of VSV pseudotyping, and studies a virus (SARS-CoV-2) that is already circulating and evolving rapidly in humans. Therefore, I would consider this work to be safe itself (due to use of VSV), and not overly risk: it is not using a potential pandemic virus (like a bat CoV), and is exploring mutations that are likely to arise naturally over the next 6-12 months of virus evolution, and whose study has direct applications for vaccine and antibody development. Nonetheless, given the controversy around other experiments that for instance have involved direct engineering of actual potential pandemic viruses (not pseudoviruses, not already circulating in humans), I think a few sentences should be added explaining the justification for this work and the reasons why the benefits would seem to outweigh the risks.

Response: We sincerely appreciate the suggestion. We have highlighted this in the revised

discussion section (Line 374-376).

- I agree with the overall points, but do think that some aspects of the interpretation are a bit too dire. The history of other rapidly evolving viruses (influenza, coronaviruses) suggests that people still make neutralizing antibodies to new strains late into their lives, probably due to the combined action of new B cells and affinity maturation. So perhaps the way things are framed could also take that into account.

Response: Thanks for the suggestion. With further affinity maturation, the variant-specific memory B cells might provide more and better NAbs against new variants. Repeated stimulation by Omicron-specific vaccines or infections of Omicron variants could also contribute to reshaping the immunity landscape. Future studies are expected to further decipher the immune imprinting. We have added discussions about this (Line 350).

- How many donors did the B cells come from, and how consistent were results among donors?

Response: Totally we collected PBMC from 50, 39 and 21 individuals with BA.1, BA.2 and BA.5 breakthrough infection, respectively, as specified in Supplementary Table 1. In addition, samples from another 15 individuals with BA.5 breakthrough infection (the last 15 individuals in Table S1) were also added to the neutralization assays in this revision, but were not subjected to antibody isolation. We pooled samples from ~20 individuals with the same immune history together as a batch for FACS analysis and scV(D)J-seq, so the results directly reflect the average behavior. FACS analysis of batches from the same immune history is highly consistent (See Extended Data Fig. 3a-b in the revised manuscript for the FACS plot of replicates).

Referee #2 (Remarks to the Author):

This study sets out to investigate the impact and driving force behind convergent mutations in Omicron BA.2 and BA.5 variants. In terms of impact they use pseudotype viruses containing Spike of BA.2 and BA.5 variants and examined how they were neutralized by (1) therapeutic antibodies, or (2) by antibodies in sera collected post dose 3 of vaccine or post breakthrough infection with BA.1, BA.2 or BA.5. This shows that variants are relatively poorly neutralized by most therapeutic mAbs and post-exposure sera. In terms of driving forces they characterize the epitope repertoire of B cell derived recombinant mAbs from vaccinees who had breakthrough infections with BA.2 and BA.5 using deep mutational scanning. This shows a skewing towards non-neutralizing epitopes, particularly after BA.5 infection compared to WT exposure. Additionally, the epitope repertoire of neutralizing Abs (nAbs) derived from post-breakthrough infection RBD-reactive B cells is relatively narrow compared to that of equivalent B cells detected post vaccination or infection with ancestral/WT/Wuhan strain. Although this is relatively clearly demonstrated, the assertion that the driving force for convergent mutations is immune imprinting comes across as circumstantial rather than directly demonstrated. While most RBD reactive B cells detected post breakthrough infection are Wuhan strain cross-reactive, there is limited evidence to show that a different pattern is observed in the absence of prior exposure to Wuhan strain. Finally, the authors investigate how BA.2 and BA.5 viruses engineered to contain combinations of the convergent substitutions are neutralized by sera from participants with breakthrough infections and find that four substitutions at positions 346, 444, 452, and 486 are sufficient to confer substantial escape.

The study presents important original results that are of importance for the design of vaccines and therapeutics

Response: Thanks for recognizing our work.

The data and methodology are of high quality and validity with the exception that there is no description of how the ACE2 binding affinity of RBD variants was measured in Fig 2b, 4 etc. In past papers the authors have used surface plasmon resonance (SPR) but here they use ELISA. This should be described and ideally validated by comparison to SPR.

Response: Actually, we did not use ELISA, but used pseudovirus-based neutralization assays to determine the relative binding affinity between hACE2 and RBD of variants. Specifically, soluble hACE2 protein is considered an inhibitor to “neutralize” the pseudovirus, just like a neutralizing antibody. The resulting IC50 can indicate the efficiency of the inhibition, thus reflecting the binding capability. The approach was also used in previous studies, and the results are generally in line with gold-standard affinity assays, including SPR (<https://doi.org/10.1016/j.chom.2022.09.002>). It’s extremely time-consuming to express and purify so many RBDs of various variants for SPR, and we are not very interested in the absolute binding affinity (K_D) of the variants but their relative

affinity compared to previous well-known variants such as ancestral strain (D614G), BA.2 and BA.5. Therefore, we used this pseudovirus-based method to determine the relative binding capability. We apologize for not stating the methodology clearly enough and have modified the corresponding text in the revised manuscript (Line 698-700).

The statistics used appear to be appropriate with the exception that in Extended Data Figs 5, 6 & 7d: it is not clear why the statistical comparisons are pairwise for adjacent exposure groups rather than comparing all to the group only exposed to WT virus?

Response: We have adjusted the Extended Data Figs. 5-7d as suggested, comparing mAbs from Omicron breakthrough infection convalescents with the mAbs from WT.

Major Comments

Abstract: is long, and lacks clarity so that it is difficult to discern the main aims and findings that: (1) nAb responses converge on a relatively restricted range of epitopes after breakthrough infection with variant strains as compared to initial exposure to WT strains, and (2) that antibodies become increasingly directed towards non-neutralizing epitopes after these breakthrough infections.

Response: Thanks for the suggestion. We have modified the Abstract in the revised manuscript. Specifically, the technological description is omitted, and the major findings are emphasized.

The statement “Additionally, the precise convergent RBD mutations and evolution trends of BA.2.75/BA.5 subvariants could be inferred by integrating the neutralization-weighted DMS profiles of mAbs from various immune histories (3051 mAbs in total).” is not a study finding.

Response: Thanks for the suggestion. We have modified the Abstract in the revised manuscript.

The statement “Moreover, we demonstrated that as few as five additional convergent mutations based on BA.5 or BA.2.75 could completely evade most plasma samples, including those from BA.5 breakthrough infection, while retaining sufficient hACE2-binding affinity.” differs from line 309 which suggests four substitutions are sufficient.

Response: We indeed found that BA.2.75 with 4 mutations could nearly escape all tested plasma samples; however, BA.5-derived mutants need more NTD mutations, which have not been deeply investigated, to fully omit the neutralization of plasma from BA.5 convalescents. Since we believe this statement is not as important as other major findings, we have omitted the sentence from the abstract. We have also unified related descriptions in the main text.

Line 141: “Similar to that reported in BA.1 breakthrough infection, immune imprinting, or so-called “original antigenic sin”, is also observed in BA.2 and BA.5 breakthrough infection 2,36-39. Post-vaccination infection with BA.2 and BA.5 mainly recalls cross-reactive memory B cells elicited by

wildtype-based vaccine, but rarely produces BA.2/BA.5 specific B cells, similar to BA.1 breakthrough infection (Fig. 3a-b). This is in marked contrast to Omicron infection without previous vaccination (Fig. 3c).”

It is not sufficient to show one example of each, please include a plot or table with the percentage of Omicron RBD reactive B cells that are Wuhan/WT RBD cross-reactive. It would also be easier to understand the cross-reactivity profile by showing FACS plots of Omicron RBD-PE/APC versus WT-RBD BV605. Additionally, there is no mention of the participants who had BA.2 infection without prior vaccination in Supplementary Table 1. How many were there, what were there ages and sampling times?

Response: The shown FACS plot is the average results of the pooled samples from dozens of individuals, instead of an example from a single participant, with the aim to reduce the uncertainty due to low cell input. Therefore, we did not perform the FACS separately; however, we provided an additional plot that shows replicates of the experiment using different sample pools from breakthrough infection (Extended Data Fig. 3a-b). Also, our results are consistent with other recently reported results on BA.1 breakthrough infection convalescents by other groups (<https://doi.org/10.1101/2022.09.21.508922>). The number of samples from convalescents without prior vaccination is too small to perform separate analyses, so all samples were pooled together for FACS. The information of these unvaccinated participants has been added to Supplementary Table 1.

Figure 3d - did the authors use separate oligo tags for BA.2/BA.5 RBD and WT RBD so the ELISA binding activity of mAbs from each B cell could be related to cross-reactivity by hashtag binding ~ flow cytometry?

Response: We indeed used the Feature Barcode technique during the immune profiling to help identify the binding specificity before the selection of mAbs for expression. As shown in the figure below, the B cells encoding cross-reactive mAbs (confirmed by ELISA) generally exhibited a higher ratio of RBD^{WT} feature barcode.

Line 560: “Cells are also labelled with biotinylated RBD conjugated to oligo SA” – which RBD’s were used – Omicron only or both Omicron and WT with different tags? What are the brands and Catalogue numbers of the oligo tags?

Response: WT RBD and Omicron RBD were both labeled with oligo-conjugated streptavidin. Omicron RBD (BA.2 or BA.5) were labeled with TotalSeq-C0971 Streptavidin (Biolegend, 405271) and TotalSeq-C0972 Streptavidin (Biolegend, 405273); WT RBD were labeled with TotalSeq-C0973 Streptavidin (Biolegend, 405275) and TotalSeq-C0974 Streptavidin (Biolegend, 405277). The above description has also been added to the Methods (Line 580-598).

Regarding Figures 3 and 4: Could the data and inferences be affected by the relatively small number of mAbs derived from post-BA.5 infection B cells? In particular, for Figure 4a it may be expected that the more mAbs used the greater the diversity of immune selection pressure. If a random narrower pool of BA.2 infection derived mAbs is used will the same range of substitutions be selected?

Response: This is an important issue. We believe that the current number of mAbs from BA.5 convalescents is sufficient for the analysis. As shown in the figure below, we randomly selected 200 mAbs from BA.2 convalescents and performed the same calculation as Fig. 4a. The random sampling was performed 8 times. Although there is some fluctuation in the absolute values of each residue, the downsampled results would not cause a substantial impact on our conclusion.

Figure 4: References to panels b/c in the legend seem to be incorrect. What does class refer to at the bottom of the IC50 data heat maps? What are the individual values (dots) in the assay for “IC50 of ACE2 against these variants”. Again, I am unclear on how this data is generated.

Response: We modified the legend of Figure 4 for clarity. Specifically, Fig. 4b used DMS profiles of all mAbs except those from vaccinated SARS convalescents (which means mAbs elicited by WT/BA.1/BA.2/BA.5), and the DMS scores of each antibody were weighted by its neutralizing activity against BA.2.75 or BA.5, as indicated in the title of the panels.

The “classes” of mAbs were defined according to the taxonomy originally proposed by Barnes et al. (<https://doi.org/10.1038/s41586-020-2852-1>), which is widely used in the field. We added this label to facilitate understanding of antibody classification, especially for those who are familiar with Barnes’s definition. We have added a description in the figure legend (Fig. 4c legend).

The dots in the “IC50 of hACE2” indicate an independent replicate of the experiment that uses soluble hACE2 as an inhibitor to evaluate the relative ACE2-binding affinity of various pseudoviruses. Five biological replicates were performed for each pseudovirus, and each replicate is shown as a dot. This is also further specified in the main text and the Methods section (Line 700).

Extended Data Fig 4b: there is no scale for the colours used to indicate IC50 levels.

Response: The color scale has been added in the revised figure.

Extended Data Fig 5a: Specify that these are results of DMS in the legend.

Response: The figure legend has been modified as suggested.

Line 159: “Among them, mAbs in groups A, B, C, D1, D2, F2, and F3 compete with ACE2 and exhibit neutralizing activity (Extended Data Fig. 5a-d, 6a-d); while mAbs in groups E1, E2.1, E2.2, E3, and F1 do not compete with ACE2 (Extended Data Fig. 7a-c).”

There seems to be nAb activity against D614G by mAbs in groups E1 and E2? I am also struggling to see which data in these figures refers to mAbs competing ACE2 binding? I can see data for how mAbs select substitutions in RBD and neutralize?

Response: Antibodies in Groups E1 and E2 indeed exhibit neutralizing activities, but their neutralization is not attributed to the direct competition with ACE2. A recent study reported a NAb in Group E1, and demonstrated that it neutralizes the virus by inhibiting membrane fusion (<https://doi.org/10.1126/sciimmunol.add5446>), which may explain the mechanism. The ACE2 competition data are shown as t-SNE plot in Fig. 3h. We have added the corresponding figure reference in the text (Line 154).

Line 171: “However, BA.5-elicited mAbs showed a more distinct distribution compared to BA.1, with a significantly increased proportion of mAbs in group D2 and E2.2, and decreased ratio of antibodies in groups B and E2.1. The main reason is that the F486 and L452 mutations carried by

BA.5 make these cross-reactive memory B cells unable to be activated and recalled (Fig. 3f, Extended Data Fig. 5b, 6a and 7a).”

This seems logical given that these are the only substitutions that distinguish BA.4/5 within the epitope group footprints, but then again there are no substitutions that would account for the relative lack of group D2 mAb elicited by BA.1 and BA.2?

Response: Actually, D2 mAbs are also rare in WT convalescents/vaccinees, and its proportion is easily affected by sampling bias. We are not very confident about the reason. Maybe the N440K near the D2 epitope harbored by all Omicron variants altered the immunogenicity. The immunogenicity drift from WT to Omicron is too strong to give a clear explanation, so in our analysis, we mainly focus on the comparison between the distribution of mAbs from BA.1/BA.2/BA.5 convalescents.

Line 252: “As expected, R493Q and N417T are not major contributors to antibody evasion, but 493Q significantly benefits ACE2 binding.” Why is it expected that N417T is not a major contributor when it is a substitution selected by antibodies in DMS analysis, and is in group A epitope region?

Response: In WT RBD, the residue 417 is a lysine (K), and both K417N and K417T escape the majority of mAbs in group A. As demonstrated in Fig. 4c, N417T did not contribute to significant antibody evasion of BA.2-effective antibodies. And the DMS analysis only reflects the selection of K417T, not N417T.

Discussion line 333: “The interaction between convergent evolution of escaping variants and less diversified antibody repertoire would ultimately lead to a highly evasive variant, posing a great challenge to current vaccines and antibody drugs” Should this statement be specified as a less diversified neutralizing antibody repertoire? I’m not sure that the study addressed the overall Ab repertoire?

Response: Thanks for pointing this problem out. Using “neutralizing antibody” here is appropriate. We have corrected this.

Minor comments

Line 485 “For convenience, Fig. 2a-c are also included here” should say Fig 3a-c.

Response: We have corrected the problem. Extended Data Fig. 3 has been rearranged.

Antigenicity drift is normally termed antigenic drift.

Response: The term has been corrected as suggested.

Referee #3 (Remarks to the Author):

Cao et al. combine sequence analysis, neutralization sensitivity, ACE-2 sensitivity, and modeling of deep mutational scanning to discern immune pressure being placed on SARS-CoV-2 and how it escapes from neutralization. Overall, it is an excellent study that makes significant points for vaccine design and for understanding SARS-CoV-2 humoral immunity. The sequence analysis in the manuscript shows the presence of shared mutations in different omicron lineages. The authors also show that monoclonal antibodies and vaccinee sera struggle to neutralize the newly emerging variants, but it should be stated in the text that most of the titers only change by 3-6 fold.

Response: The description has been modified (Line 108-122).

The authors then isolate ~1700 monoclonal antibodies and show how the antibody repertoire shifts in epitope specificity after omicron infection and that many of the responding B cells are cross-reactive with earlier Wuhan-Hu-1 sequence RBD . The authors then use deep mutational analyses coupled with neutralizing activity of the mAbs to identify sites that could mutate to escape the neutralizing antibodies. These potential mutational sites could be useful for defining the next variants to emerge. Generating pseudoviruses with the potential mutation combinations resulted viruses that fully or nearly completely escaped vaccinee sera. In total, this work would indicate SARS-CoV-2 can completely escape from vaccine antibody neutralization responses without compromising ACE-2 binding. The concept of original antigenic sin or antibody imprinting is of concern and the authors show at the monoclonal antibody level how the humoral response to BA.1, BA.2, and BA.5 infection shifts relative to what happens with Wuhan-Hu-1 infection. This shift in epitopes recognized by omicron-induced antibodies is problematic based on the authors demonstration that the sublineages are continuing to accumulate escape mutations and broader epitope specificity will be needed to maintain neutralization activity against the newly emerging subvariants. Cao and colleagues define plausible mutations that could accumulate in SARS-CoV-2 and show these viruses are completely resistant to neutralization by vaccines sera. Overall, the references and statistical analyses seem appropriate. The results support the conclusion. Instances where the authors speculate on future events are noted below and suggested to be moved to the discussion.

Response: We have moved the speculation in the result section to the discussion in the revised manuscript, as suggested.

Major comments:

Figure 1: The first point made by the authors is that Omicron sublineages contain the same mutations suggesting convergent mutational patterns in Spike relative to the whole genome. Meaning that while viral genomes place subvariants into different clades in a phylogenetic tree, the spike sequences include several common amino acid changes. How significant is this result? This

reviewer believes this observation has been seen with earlier variants from completely different PANGO lineages (for example E484K, L452R).

Response: The current convergent evolution of RBD is highly significant and unprecedented, with hundreds of designated Omicron sublineages carrying those convergent mutations. Although variants before Omicron could also share some immune-evasive mutations, such as K417N/T, L452R, E484Q/K, we did not observe such significant and rapid appearance of multiple sublineages carrying similar mutations. For example, only a very small proportion of Delta sublineages (AY.*) evolved K417N/T and E484K/Q, and nearly no Beta/Gamma sublineages evolved L452R. Actually, K417/E484 mutations were simply the best choice for the virus to escape NABs elicited by WT infection/vaccination at that time, as shown in Extended Data Fig. 8a, but many other mutations may also exhibit an advantage. The scenario is different now, where the possible selection of RBD mutations for the virus is very limited and concentrated, promoting the convergent evolution.

1. A phylogenetic tree of spike sequences from each of the viruses shown in 1a would help the reader understand the extent of convergent spike sequences that come from different sublineages.

Response: Thanks for the suggestion. We have tried to construct a phylogenetic tree using the Spike sequences only to facilitate the understanding of the convergent evolution of RBD, as shown below. It indeed showed some mixing of convergent variants among different lineages, which reflects the convergent phenomenon, but not significant enough.

We have substantially rearranged Fig. 1, and included some lineage analysis. We believe current Fig. 1 would present the convergent phenomenon much better than before, and also better than the phylogenetic tree using the Spike sequences only.

Genome_Nextstrain

Spike_Nextstrain

2. If one posits that the sublineages are evolving in the same geographic region, then immune pressure may account for similar spike mutations. Have you examined the presence of these mutations by geographic location?

Response: This is indeed an interesting topic. We previously showed plasma from Delta breakthrough infection convalescents showed higher resistance against the evasion of BA.4/5 which carries L452R (<https://doi.org/10.1016/j.chom.2022.09.018>). There is also an observation that variants sequenced in India seem to have less L452R mutations compared to many other geographic regions. This could be attributed to the previous prevalence of Delta VOC in India, and suggests that indeed region-specific immune pressure may shape the region-specific Spike mutations. However, this analysis is beyond the specialties of our group and out of the scope of this manuscript. This is expected to be investigated by more professional epidemiologists other than us.

3. How many sublineages in BA.4/5 must have a mutation observed in BA.2 for the mutation to be considered shared with BA.2? Is finding the mutation in one BA.4/5 sequence considered convergent evolution. Providing a threshold would help the reader.

Response: Currently, there is no standard to define a threshold for a particular convergent mutation. However, the observed convergence on these hotspots, including R346, K356, L452, K444, V445, N450, N460, F486, F490 is too significant to ignore. In this paper, we choose these sites as convergent mutation sites since we have observed at least more than 5 independent emergences in distinct lineages of BA.2 and BA.5 that exhibited growth advantage. The precise definition of the threshold should be more appropriate to be proposed by experts on epidemiology other than us. We have added related descriptions in the main text (Line 242)

4. Is convergent evolution only present in BA.2, BA.2.75, BA.5? If you compared Delta and Kappa sublineages would you see the same phenomenon?

Response: The number of Kappa sublineages is too small to identify any convergent evolution. As for Delta sublineages (AY.*), the majority of them were defined by mutations out of Spike and the convergent evolution of different sublineages with similar Spike mutations are much rarer compared to the current situation.

Figure 2:

While binding affinity and neutralization inhibitory concentration are correlated they are not the same parameter. Please replace ACE2 binding affinity with ACE2 inhibition or ACE2 inhibitory concentration.

Response: Thanks for the suggestion. We used “IC50 of hACE2” in the figures, and the corresponding description throughout the main text is also revised, using the term “IC50 of hACE2”, “inhibitory efficiency of hACE2”, or “hACE2-binding capability”.

1. Line 76 mentions BA.5.5.1 being resistant, but I think BA.5.1.12 was meant to be mentioned since it is more resistant than BA.5.5.1.

Response: Thanks for pointing this out. It was an error in the submitted version. We have rewritten the description of Fig. 2 in the revised manuscript.

2. Line 110: Please delete “they may tolerate even more escape mutations” as this statement is not shown with data and is only speculation.

Response: We have deleted this statement as suggested and rewritten the description of Fig. 2 in the revised manuscript.

3. Lines 126-130 are interpretation of the results and would be more appropriate in the discussion.

Response: We have modified these sentences and moved them to the discussion as suggested.

Figure 3:

Could you speculate in the discussion on whether germinal center B cells in lymph nodes would have the same cross-reactivity and epitope shift. Could germinal center B cells that are responding to omicron infection or vaccination show different specificity and mutation percentages. It is plausible that at the time you looked the new antibody responses was still ongoing.

Response: We agree that at the time of sample collection, the GCs are still active and B cells are evolving for affinity maturation. The further matured mAbs could achieve higher neutralizing potency and breadth, but their targeting epitopes should not change a lot. Nevertheless, the waning immunity will make the serum neutralization titers continuously decrease, and a booster is needed to stimulate the matured memory B cells. These results were shown by a recent report that investigated the long-term (5-6 months after infection) BA.1 breakthrough infection convalescents. We have discussed this in the revised manuscript (Line 347-352).

Figure 3d: From how many BA.2 and BA.5 convalescent individuals were the antibodies isolated.

Response: We collected PBMCs from a total of 50, 39 and 21 individuals with BA.1, BA.2 and BA.5 breakthrough infection, respectively, as specified in Supplementary Table 1. In addition, samples from another 15 individuals with BA.5 breakthrough infection (the last 15 individuals in the table) were also added to the neutralization assays in this revision, but were not subjected to

antibody isolation. We pooled samples from ~20 individuals with the same immune history together as a batch for FACS analysis and scV(D)J-seq.

Minor comments:

Figure 2a, 5b: Define what the symbol in the table means. I assume it means no neutralization.

Response: “*” means IC50 is larger than the limit of detection (> 10µg/mL), which is specified in the corresponding figure legends. We also labeled it in the revised figures.

Figure 2a: The color scheme will be difficult for red-green color blind readers. It may also be better to color squares with titers between 100-1000 since white usually means no neutralization.

Response: Thanks for the suggestion. We modified the brightness of the red used for Fig. 2a and Extended Data Fig. 2a to make it more friendly for color-blind readers, as confirmed by <https://davidmathlogic.com/colorblind/#%23FFFFFF-%23E2EFDA-%23FFADAD-%23FFFFFF> (See also the figure below).

However, as the white background was consistently used in our previous report (<https://doi.org/10.1038/s41586-022-04980-y>), and also in our posts on Twitter and bioRxiv, which has been widely accepted, we would like to keep the white background for IC50 between 100-1000 to avoid confusion.

Figure 4c: labels indicating virus is shown in rows and antibodies are shown in columns would be helpful.

Response: Thanks for the suggestion. We added these labels for Fig. 4c and Extended Data Fig. 9a.

Line 346: I do not think the use of revoke conveys the intended meaning of the sentence. Perhaps, invoke, recall, engage, or re-stimulate would be better word choices.

Response: Thanks for the correction. However, we decided to delete this paragraph because of limited article length.

Reviewer Reports on the First Revision:

Referees' comments:

Referee #1 (Remarks to the Author):

The authors have satisfactorily addressed the comments.

Referee #2 (Remarks to the Author):

The authors have done a fantastic job to address the issues raised. The abstract is now much more concise and emphasizes the important findings of this study. I appreciate the authors efforts to clarify some of the methods that I was unsure about.

Referee #3 (Remarks to the Author):

The revised manuscript is much improved. The authors have responded to most of the critiques in a satisfactory way. The only critique this reviewer believes is still unanswered and affects the scientific rigor of the work is the definition of convergent evolution in Spike. The authors say there is no threshold for calling a mutation convergent, yet that is one of the main points of the manuscript--the word is in the title. Figure 1 has been revised to show the number of convergent mutations. How did you decide which mutations are convergent? It seems you are showing mutations that are present in an undefined percentage of omicron sub lineages. While there are appropriate evolutionary biologist mathematical models to show the presence of a genetic marker is converging among different species, at least here you could state that you called mutations found in X% of Omicron sub lineages to be convergent. Saying that the convergence is "too significant to ignore", but other people will define what is convergent does not seem like the right approach here. This is a problem that can be easily solved before this work is published in Nature by defining some criteria for how you focused on certain mutations. it will also help you not focus on mutations that are not meeting those criteria. The criteria can also guide subsequent studies that will examine the next sets of sub lineages that emerge.

Author Rebuttals to First Revision:

Response to the reviewers

Reviewer #1 (Remarks to the Author):

The authors have satisfactorily addressed the comments.

Response: Thank you again for your comments.

Reviewer #2 (Remarks to the Author):

The authors have done a fantastic job to address the issues raised. The abstract is now much more concise and emphasizes the important findings of this study. I appreciate the authors efforts to clarify some of the methods that I was unsure about.

Response: Thank you for your suggestions which helped substantially improve our manuscript.

Reviewer #3 (Remarks to the Author):

The revised manuscript is much improved. The authors have responded to most of the critiques in a satisfactory way. The only critique this reviewer believes is still unanswered and affects the scientific rigor of the work is the definition of convergent evolution in Spike. The authors say there is no threshold for calling a mutation convergent, yet that is one of the main points of the manuscript- the word is in the title. Figure 1 has been revised to show the number of convergent mutations. How did you decide which mutations are convergent? It seems you are showing mutations that are present in an undefined percentage of omicron sub lineages. While there are appropriate evolutionary biologist mathematical models to show the presence of a genetic marker is converging among different species, at least here you could state that you called mutations found in X% of Omicron sub lineages to be convergent. Saying that the convergence is "too significant to ignore", but other people will define what is convergent does not seem like the right approach here. This is a problem that can be easily solved before this work is published in Nature by defining some criteria for how you focused on certain mutations. it will also help you not focus on mutations that are not meeting those criteria. The criteria can also guide subsequent studies that will examine the next sets of sub lineages that emerge.

Response: Thanks for the suggestion. We agree with the importance of a definition of convergent evolution and convergent mutations in this paper, which was also recommended by Reviewer #3. As mentioned in the previous response to Reviewer #3, we consider a residue on RBD to be a

convergent site if the mutations on it were observed in at least 5 independent lineages, which exhibited growth advantages. In this revision, we have visualized this definition in the newly added Extended Data Fig. 1, added a Method section to explain this in detail, and specified this definition in the Introduction (Line 61).

We also added some sentences at the beginning of the Discussion (Line 338). Previously, N501Y was considered a convergent mutation, which could be attributed to the enhancement of ACE2-binding affinity. K417 and E484 were also considered to exhibit some convergence patterns, but the convergent evolution on these sites was not so significant and rapid as recently observed. We believe the current Introduction and Discussion sections have further explained the concept of convergent evolution, and the new Extended Data Fig. 1 will help the readers understand the significance of the recent convergent evolution of SARS-CoV-2 RBD.